

# Flat bands and compact localised states: A Carrollian roadmap

Nisa Ara[1*], Aritra Banerjee[2†], Rudranil Basu[1‡] and Bhagya Krishnan[1∘]

**1** Department of Physics, Birla Institute of Technology
and Science Pilani, Zuarinagar, Goa 403726, India
**2** Birla Institute of Technology and Science, Pilani Campus, Rajasthan 333031, India

★ p20210035@goa.bits-pilani.ac.in , † aritra.banerjee@pilani.bits-pilani.ac.in ,
‡ rudranilb@goa.bits-pilani.ac.in , ∘ p20190007@goa.bits-pilani.ac.in

## Abstract

We show how Carrollian symmetries become important in the construction of one-dimensional fermionic systems with all flat-band spectra from first principles. The key ingredient of this construction is the identification of Compact Localised States (CLSs), which appear naturally by demanding *supertranslation* invariance of the system. We use CLS basis states, with inherent *ultra-local* correlations, to write down an interacting theory which shows a non-trivial phase structure and an emergent Carroll conformal symmetry at the gapless points. We analyze this theory in detail for both zero and finite chemical potential.



# 1   Introduction

The theory of strongly correlated electrons has reigned over much of theoretical physics for the last many decades. A quantum field theoretical study of such systems has brought about many useful techniques and sharp insights into physically realizable quantum systems (See [1] for example). It is now a well-known phenomenon that the in-depth understanding of local correlations in electron motion is the driving force that leads to emergent electronic properties such as superconductivity or (various forms) of magnetism. As correlations change, one could expect that new topological phases can be induced, and more universal phenomena can be uncovered. A large volume of work in theoretical physics has therefore been focused on studying non-trivial topological and geometric properties of quantum systems, especially the fermionic ones.

In this connection, one of the most intriguing situations for lattice theories is the one where one has constant energy independent of momenta, i.e. a flat dispersion [2–4]. These situations can occur when the group velocity of a localised excitation can vanish everywhere due to destructive interference on a Bloch band. One can also equivalently think of this as an infinite effective mass limit for the excitations. Flat dispersions, although seemingly trivial from a dynamics point of view, have recently attracted researchers working on a vast spectrum of systems and phenomena, ranging from Moiré patterns in multi-layer graphene [5–7], superconductivity [8,9], fractional Quantum Hall Effect [10], non-Hermitian quantum systems [11] and even in the dynamics of shallow water [12]. In Moire systems, flat bands originate due to intricate periodic boundary conditions, rendering the free Hamiltonian kernel spectrum trivial [7]. On the other hand, in the absence of Moiré geometry and irrespective of boundary conditions, flat bands may arise, as mentioned earlier, due to precise destructive interference between sub-lattice Bloch waves. As a result, despite having finite hopping amplitudes, local excitations don't propagate and remain confined within a few local states. These are known as *Compact Localized States* (CLS) [13,14], i.e. states that decay outside a finite support region.

Fueled by many real-life applications, the design of such CLS, and consequently of flat bands, has been a focus of theoreticians over the last few years [15]. At the level of single particle spectrum, flat bands imply the existence of a large number of degenerate states, and hence a large number of generators of, possibly continuous, symmetries. Various different approaches towards the generation of CLS exist in the literature, including origami rules [16], band engineering [17–19] and block partitioning of Hamiltonian systems based on graph theory [20]. Despite this, a systematic study of these symmetries in terms of Lie groups/ algebras has been scarce. This is where the current work comes into play.

In [21], flat bands appearing in bilayer Moiré graphene and generic ladder systems were shown to be a result of the existence of an infinite number of conserved quantities corresponding to global space-time symmetries, aka *supertranslations*. One could show the invariance under this supertranslation symmetry effectively makes Hamiltonian densities at two spatial points commute. This further points towards the emergence of a *Carrollian* symmetry in such a system, which arises in the speed of light going to zero ($c \to 0$) limit of the relativistic Poincaré symmetry [22, 23]. Evidently, Carroll symmetries can be realised geometrically on space-time manifolds with degenerate metrics, with one null direction [24–26]. Consequently, since supertranslations are just angular direction-dependent translations along a null line, it is natural to expect them to be part of the generators of Carroll algebra and the conformal cousin thereof.[1]

Carrollian theories actually have a way of serendipitously emerging in various intriguing physical situations. The importance of Carroll and conformal Carroll invariant theories came into play since conformal Carroll algebra (CCA) can be shown to be isomorphic to the Bondi-Metzner-Sachs (BMS) [27,28] algebra in one more dimension. The BMS algebra, on the other hand, is the asymptotic symmetry algebra of asymptotically flat spacetimes, making Carrollian dynamics important in the physics of scattering in flat space, and consequently in *Flat Space Holography* [29–35].[2] But not only in the context of quantum gravity, Carrollian symmetries seem to appear in various accessible physical situations, including hydrodynamics of the quark-gluon plasma [40,41], in cosmology [42] and on the event horizons of generic black holes [43] as well as on null hypersurfaces [44].

An intuitive association of flat bands with Carroll symmetries appears in massless Dirac fermions. As far as Lorentz group invariance is concerned, one naturally associates the fermi velocity $v_F$, in this case, as an equivalent of the speed of light $c$. This $v_F$ characterizes the Dirac cone's conical angle in momentum space. If by tuning parameters, just as in the case of magic angle bilayer graphene, one can reach a situation of $v_F \to 0$, one gets a zero energy flat band, with the Dirac cone's conical angle being $\pi/2$. In terms of the Lorentz group, or rather the Poincaré one, this is tantamount to taking a $c \to 0$ Inönü-Wigner contraction. By definition, the resulting Lie algebra is the Carrollian one. It is already intriguing to realise such *ultra-local* symmetries, which one may envision to appear at ultra-high-energy situations, lead a quantum system to have flat bands. In this work, we proceed further by directly showing the emergence of CLS itself and, hence, manifest flat bands as one of the consequences of the supertranslation symmetry. Moreover, guided by these symmetries, we even construct interaction terms, which are also manifestly ultra-local.

In the recent past, the effect of strong interactions on fermionic theories whose free part is dispersionless has been studied extensively [45–50]. However, the presence of interactions significantly changes the situation. Most of the curious features, like zero-correlation length and Aharonov-Bohm caging, are expected to be destroyed if the interaction is not carefully designed using the CLS as the building blocks. For example, let's consider a single particle CLS in the $n$th flat-dispersion band generated by a fermionic mode $\alpha_p^{(n)} = \sum_{j,\gamma} C_{pj}^{\gamma} c_j^{\gamma}$, where $c_j^{\gamma}$ are site-local fermions in the $\gamma$ sublattice. The sum runs over a few of the neighbouring sites as well as sublattices. Supertranslation symmetry, and hence all the intriguing features, like ultra-locality of the model, would cease to exist if one introduces a local interaction term in the form $\sum_{\gamma,\gamma',j} D^{\gamma\gamma'} n_j^{\gamma} n_j^{\gamma'}$ or $\sum_{\gamma,j} n_j^{\gamma} n_{j+1}^{\gamma}$ for $n_j = c_j^{\dagger} c_j$. One such consequence, for example, is the variation of the gap [45] or subsystem entanglement entropy with magnetic flux in an Aharonov-Bohm set-up.

---

[1]See a detailed discussion about generators of infinite dimensional Carroll Conformal Algebra (CCA) in the Appendix.(A).

[2]See also [36,37] for a non-exhaustive list of introduction to Carrollian approach to *Celestial* holography [38,39].

Instead, we devise explicitly supertranslation invariant interaction terms to be added to the free theory, which leads to even more exciting physics, both in the case of half-filling and beyond. These particular interactions, made out of site-local CLS modes, keep the theory perfectly solvable, but the parameter space shows intriguing Quantum Phase Transitions. Particularly, this leads to an *exotic* phase with a highly degenerate ground state manifold. This requires one to find a consistent choice for a vacuum state that is, in general, a momentum eigenstate. Moreover, it can be shown that manifest ultra-locality is unfazed in this exotic phase even when a Carroll-breaking perturbation is added to the system.

The rest of the paper is arranged in the following way: In section (2) we introduce a symmetry mandated way to design ultra-local fermions, and consequently flat dispersions, in the context of a lattice system. In section (3) we will elucidate on the generic quantum mechanical structure of ultra-local flat-band theories which manifestly follows from the emergent Carroll symmetry. In section (4) we briefly discuss the introduction of explicitly supertranslation invariant four-fermion interaction terms. Section (5) and (6) delve into the quantum phase structure of interacting supertranslation invariant lattice systems, both with the filling factor constrained at half and unconstrained. Finally in (7) we conclude with a summary and clear up the road ahead towards the usage of symmetry principles in engineering flat-band systems, and their direct applications.

## 2 Carroll symmetries and flat bands

### 2.1 Ultra-local fermions in one dimension

We start with a generic free theory of two-component fermions on a one-dimensional lattice with nearest neighbour hopping (h.c. denotes a hermitian conjugate term):

$$H = \sum \mathcal{H}_j, \quad \text{where} \quad \mathcal{H}_j = \psi^\dagger_{j+1} q \psi_j + \text{h.c.}, \tag{1}$$

where $\mathcal{H}_j$ are the discrete version of Hamiltonian density and $q$ is a 2×2 matrix. With the usual canonical anticommutation structure for the fermions: $\{\psi_{i,\alpha}, \psi^\dagger_{j,\beta}\} = \delta_{ij}\delta^{\alpha\beta}$ etc. ($\alpha, \beta = 1, 2$ denote two components of the fermions $\psi$) one readily gets the following Heisenberg equation of motion:

$$\dot{\psi}_j = -i\left(q\psi_{j-1} + q^\dagger\psi_{j+1}\right). \tag{2}$$

For an arbitrary choice of $q$, this free theory predicts the time evolution of an initially localized state should spread across further lattice points as expected for finite correlation length models. Also, the effective continuum theory describing fluctuations around the ground state, in this case, should contain spatial derivatives. However, taking a time derivative of (2), we note that if $q$ is nilpotent then,

$$\ddot{\psi}_j = -\left\{q, q^\dagger\right\}\psi_j. \tag{3}$$

This is further simplified because for any nilpotent $q$, there exists a $\kappa \in \mathbb{R}$, such that the anticommutator $\{q, q^\dagger\} = \kappa^2 \mathbf{1}$. Equation (3) indicates that the stationary states are localized in space. Hence, each point in the space of nilpotent 2-dimensional matrices does correspond to a two-band *ultra-local* theory.[3] Since the odd derivatives in time like in (2) would still involve a linear combination of the two nearest neighbouring sites, the stationary states will be linear combinations of states localized at most at two neighbouring sites. These, as introduced earlier, are well-known in the literature as Compact Localized States (CLS) [13, 52–55][4] that

---

[3]Referring to our discussion of Carroll symmetries and flat bands, one could think of these nilpotent matrices as degenerate representations of the Carrollian Clifford algebra [21, 51].

[4]CLS has been experimentally observed in photonic lattices, see for example [56].

arise from destructive quantum interference. In the following, we will provide a systematic route to find the CLS in terms of the local states.

One of the most remarkable properties of the Hamiltonian (1) owing to the nilpotency of $q$ is that the Hamiltonian densities at arbitrary points commute:

$$[\mathcal{H}_i, \mathcal{H}_j] = 0. \tag{4}$$

Hence, for any lattice function $f$, the operator:

$$Q_f = \sum_j f_j \mathcal{H}_j \tag{5}$$

is a conserved charge. One can construct as many of such linearly independent charges as the number of lattice points. For arbitrary functions $f, g$, these charges also mutually commute: $[Q_f, Q_g] = 0$. To find the symmetries generated by the above charges, we define the transformation:

$$\delta_f \psi_j := -i \left[ Q_f, \psi_j \right], \tag{6}$$

where the commutator, at the level of operators, means commutator with each component of the spinor $\psi_j$. Using fundamental canonical relations and the nilpotency of $q$ together with the equation of motion (2), we arrive at:

$$\delta_f \xi_j = f_j \dot{\xi}_j, \quad \text{where} \quad \xi_j = \frac{1}{\kappa} \left( q \psi_j + q^\dagger \psi_{j+1} \right). \tag{7}$$

The new spinors $\xi$ are the ones to produce CLS at the level of single-particle states. Strikingly, the transformation (7) is known as *supertranslation* transformation in Carrollian QFT literature [57,58]. The normalization factor $\kappa = \sqrt{\{q, q^\dagger\}}$ was used in the definition of the CLS $\xi$ in order to make the transformation $\psi \to \xi$ canonical. One can of course invert the above definition and express original site local fermions $\psi$ as:

$$\psi_j = \frac{1}{\kappa} \left( q \xi_{j-1} + q^\dagger \xi_j \right), \tag{8}$$

and rewrite the Hamiltonian (1) in the form:

$$H = \sum_j \xi_j^\dagger \left( q + q^\dagger \right) \xi_j. \tag{9}$$

When expressed in terms of CLSs, the Hamiltonian clearly is a model of two non-dispersive or flat bands. To see this, let's parameterize the space of two-dimensional nilpotent matrices by two complex numbers $(\tau, \alpha)$ as:

$$q = \tau \begin{pmatrix} 1 & \alpha \\ -1/\alpha & -1 \end{pmatrix}. \tag{10}$$

For this parameterization (10), diagonalizing the site local Hamiltonian Kernel $(q + q^\dagger)$, we find that the eigenvalues, i.e. band energies, are given by $\pm|\tau|(|\alpha| + 1/|\alpha|)$. Irrespective of the values of $\tau, \alpha$, there are always a couple of edge-localized zero-energy modes for open boundary conditions, which are decoupled from the energy bands. It is easy to see that the space of this Hamiltonian Kernel is the $GL(2, \mathbb{C})$ orbit under conjugate action on the ray of upper triangular matrices in the two-dimensional representation.

One can generalize this to a wider class of Hamiltonians, which describe hopping further than the nearest neighbour. If we want to define our CLS via the localization criterion in a way analogous to (3), i.e. ultra-locality through dynamics up to the second derivative in time, we

should be careful about the nilpotency degree to be equal to 2. In that case, one needs more than one such $q$ matrix. Let $q_m$ denote $m$th nearest neighbour; then all the above arguments go through if:

$$\{q_m, q_n\} = 0, \ \{q_m, q_n^\dagger\} \sim \delta_{mn}. \tag{11}$$

But this, again, is the algebra of fermion operators. The obvious finite-dimensional representations of $P$ such matrices are $2^P$ dimensional. However, one can also choose to work with smaller ($D < 2^P$) dimensional irreducible representations. That leaves us to work with $D$ component fermions.

## 2.2 Lattice fermion generalities: Symmetries and energy scales

It is well understood that energy scales play a crucial role in the space-time symmetries, which an effective quantum field theory (QFT) enjoys. The easiest example, probably, is a Poincaré invariant scalar field theory with a marginal coupling. Let the theory be massless and conformal at a particular energy scale. However, as one goes down the energy scale and flows towards the IR, mass is generated at one-loop and conformal symmetry is broken. This reduces the global space-time symmetry group.

In the present context of free fermions on a lattice, even more dramatic scenarios can arise as far as symmetries from space-time are concerned. For example, for a simple nearest neighbour model of hopping in one spatial dimension, the dispersion is gapless and at half filling (i.e. ground state), the effective continuum description is that of left and right moving relativistic fermions [59]. Hence, the system enjoys the infinite dimensional group of conformal symmetries. However, fluctuations around the vacuum (an excited state) are described by the Schrödinger field theory and the symmetries are expressed through the Schrödinger algebra [60–62], a conformal extension of Galilean algebra.

For a gapped model like SSH chain [63, 64], again, the effective theory at half-filling. i.e. around the ground state, with the Fermi energy lying in the gap, the space-time symmetries enjoyed by the model are described by the Poincaré group. In this case, the characteristic velocity is set by the Fermi velocity $v_F$. However, the theory around the vacuum, with the chiral symmetry spoiled, is again that of the Schrödinger type. This turns out to be the universal description for fermions on a 1d lattice [59, 65].

To elucidate on the physics as described above, we will consider briefly another two-component free fermion model defined by the Hamiltonian on a lattice since this will be directly connected with flat band models described by using nilpotent matrices as in the last section:

$$H = \sum_j \left( t_1 \left( c_{j+1}^\dagger c_j - d_{j+1}^\dagger d_j \right) + t_2 \left( c_{j+1}^\dagger d_j - d_{j+1}^\dagger c_j \right) + \text{h.c.} \right). \tag{12}$$

The Fourier space Hamiltonian kernel is given by: $\mathcal{H}_k = 2t_1 \cos k \, \sigma_3 - 2t_2 \sin k \, \sigma_2$. The vector $2(t_1 \cos k, t_2 \sin k)$ has non-trivial winding $\pm 1$ in $\mathbb{R}^2 / \{0\}$. The sign of the winding depends upon the relative sign of the parameters $t_1, t_2$. The dispersion bands given by this Hamiltonian are:

$$E_\pm = \pm \sqrt{2} \sqrt{t_1^2 + t_2^2 + (t_1^2 - t_2^2) \cos(2ka)},$$

for the lattice constant $a$. If we choose $t_1 > t_2$ and focus on the valence band $E_-$, then the vacuum is described by the point $k = 0$. Expanding, for small $k$, we get:

$$E_-(k) \approx -2\sqrt{t_1^2 - (t_1^2 - t_2^2)(ka)^2} \approx -2t_1 + \frac{k^2}{2m}, \quad \text{for} \quad m = \frac{t_1}{2 \left( t_1^2 - t_2^2 \right) a^2}. \tag{13}$$

This is the dispersion relation of the non-relativistic Schrödinger field, as expected. On the other hand, dynamics around the ground state, i.e. the half-filled scenario ($k = \pi/(2a)$), we

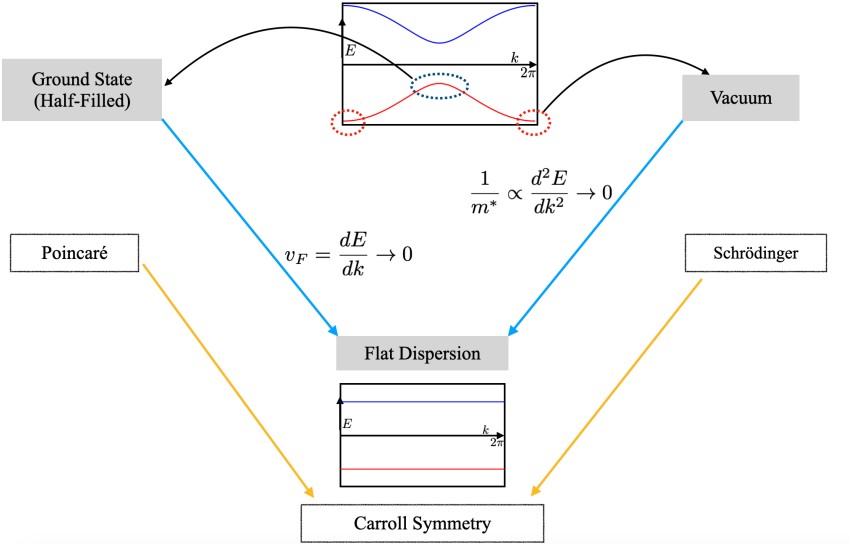

Figure 1: Schematic to show different pathways to flat dispersion relations (and Carroll symmetries) for lattice fermions in different configurations.

have for the same valence band, a dispersion relation analogue to the relativistic one:

$$E_-(k') \approx -\sqrt{m'^2 v_F^4 + k'^2 v_F^2}\,. \tag{14}$$

Here the Fermi velocity is given by $v_F = 2a\sqrt{t_1^2 - t_2^2}$ and the effective mass is $m' = \frac{t_2}{2(t_1^2 - t_2^2)a^2}$. Hence, in the continuum description, fluctuations around the half-filled state are Poincaré invariant, with the velocity scale set by $v_F$. The difference between the dispersion and, therefore, the symmetry groups of the vacuum and the ground state (see Fig.(1)) occurs due to the sign and/or reality of mass depending on values of $t_{1,2}$, i.e. unitarity of the system.[5]

Now, the bandwidth of the bands vanishes at $t_2 \to t_1$, the specific limit we are interested in. This is known as the Creutz ladder (Fig. (2a)) in the literature [66]. As $t_1 \to t_2$ in the first description (13), the vacuum becomes infinitely massive, and fluctuations become ultra-localized. On the other hand, for fluctuations around the ground state, as in the second description above (14), the Fermi velocity identically vanishes, and the mass diverges. However, it does so, keeping the combination $m'^2 v_F^4$ a finite constant. This effectively pushes both descriptions to a dispersionless (or flat) point, or in other words, enforces both the high energy (14) as well as the low energy (13) scale physics to collapse to a single energy scale of the flat band. For more details on the interplay between energy scales and symmetry groups for Carroll invariant theories, particularly as one flows toward the IR in a Wilsonian way, the interested reader may refer to [57,67], and to [68], specifically for a discussion on UV/IR scale mixing for Carrollian theories.

In fact, in this limit, the model (12) becomes equivalent to the one we generated using nilpotent matrices in (1) with the identification of $\tau = 2t_1$ and $\alpha = 1$ for the parameterization (10). Intriguingly, in this limit, both the Schrödinger symmetry for the higher energy vacuum state and the Poincaré symmetry of the ground state go through two different transformations to give rise to identical Carroll symmetry algebras. While for the Poincaré case, this process is given by a well-known Inönü-Wigner contraction [69], the Schrödinger symmetry

---

[5]Curiously, for Lorentzian signature and real value of mass, we have to expand the square root in the vacuum case. For the real value of $v_F$ or effective mass $m'$, we are not expanding the square root of the dispersion in the ground state.

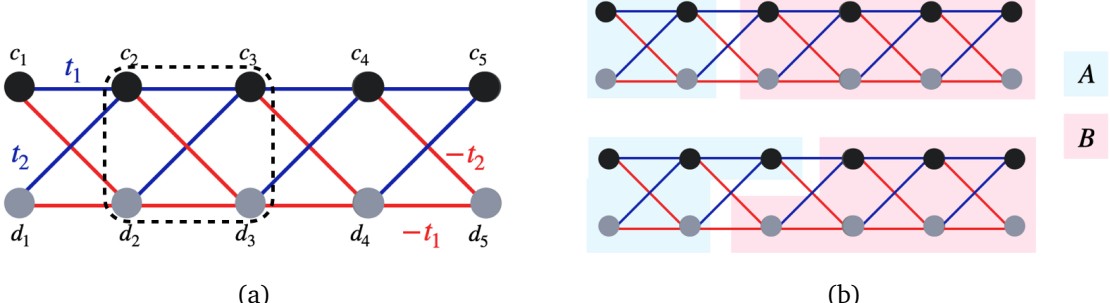

Figure 2: (a) The ladder geometry with hopping terms as per (12). The dotted region marks four site local modes constituting a CLS; (b) shows the entanglement bipartition cut, which divides the ladder into two halves. The two cuts show closed and open boundary conditions.

goes through an infinite mass limit[6] on the respective commutator [70,71]. See Fig. (1) for a summary of this argument.

From the point of view of single particle dispersion, the energy band is completely independent of momentum $k$ as $t_2 \to t_1$. The real-space continuum theory at this point is devoid of any spatial derivatives. As a result, at this precise point, supertranslations emerge as a new and infinite number of symmetry generators in the system, thus enlarging the parent symmetry groups to an infinite-dimensional one despite the absence of conformal symmetry. Note again that this dynamics at the very point with $t_2 \to t_1$ limit can be intrinsically generated using the nilpotent matrices (1), making our construction of flat dispersion models simple yet profound.

## 3 Consequences of ultra-locality: Dynamical features

Now that we have introduced generic two-band ultralocal fermions via nilpotent matrices, we would like to characterise the quantum mechanical consequences of such a model. In order to proceed further with this, we choose the model (12) with our region of interest $t_1 = t_2$, i.e. the flat band limit, which in terms of the parameterization (10) of nilpotent matrices correspond to $\alpha = 1$ and $\tau = t_1 \in \mathbb{R}$.

### 3.1 Correlation functions and entanglement

With the parameterization described above, and using (8), the CLS modes $\xi^\dagger = (\alpha^\dagger \ \beta^\dagger)$ can be related to the site-local oscillators $\psi^\dagger = (c^\dagger \ d^\dagger)$ in (12) via the combinations:

$$\alpha_j = \frac{1}{2}\left(c_j + d_j - c_{j+1} + d_{j+1}\right), \qquad \beta_j = \frac{1}{2}\left(c_j + d_j + c_{j+1} - d_{j+1}\right). \tag{15}$$

In terms of the CLS, the Hamiltonian (12) takes the simple form:

$$H = 2\tau \sum_j \left(\alpha_j^\dagger \alpha_j - \beta_j^\dagger \beta_j\right). \tag{16}$$

It is now evident that for $\tau > 0$, the ground state is the one filled by all the $\beta$ modes. The ultra-locality of the CLS in this ground state implies:

$$\langle \beta_i^\dagger(t)\beta_j(0)\rangle = e^{-2i\tau t}\delta_{i,j}, \ \langle \alpha_i^\dagger(t)\beta_j(0)\rangle = 0 = \langle \alpha_i^\dagger(t)\alpha_j(0)\rangle. \tag{17}$$

---

[6]Note that an infinite mass limit on the continuum Schrödinger action leads to the Carroll fermion action $\sim \int d^2x \ \psi^\dagger \partial_t \psi$, where only the time derivative term survives.

This is also reflected in the ultra-locality in time-dependent two-point functions of site-local modes:

$$\langle c_i^\dagger(t) c_j(0) \rangle = \left( \frac{1}{2} \delta_{i,j} + \frac{1}{4} \delta_{i,j+1} + \frac{1}{4} \delta_{i,j-1} \right) e^{-2i\tau t},$$

$$\langle c_i^\dagger(t) d_j(0) \rangle = \frac{1}{4} \left( \delta_{i,j+1} - \delta_{i,j-1} \right) e^{-2i\tau t}, \qquad (18)$$

$$\langle d_i^\dagger(t) d_j(0) \rangle = \left( \frac{1}{2} \delta_{i,j} - \frac{1}{4} \delta_{i,j+1} - \frac{1}{4} \delta_{i,j-1} \right) e^{-2i\tau t}.$$

This has a strictly zero-correlation length beyond a plaquette of 4 sites as per Fig. (2a). This correlation is a typical feature of the so-called electric-type Carrollian theories in the continuum [42,72], where time derivatives dominate the action.

In fact, the above ultra-locality is a direct consequence of the supertranslation charges $Q_f$ (5). To see that, let us consider the two-point function $G(t_1, x_1; t_2, x_2)$ in the ground state of such a theory, assumed to be invariant under supertranslation.[7] Time translation and (lattice) spatial translation symmetry impose the two-point function to be of the form $\tilde{G}(t_1 - t_2, x_1 - x_2)$. We make a further assumption, taking advantage of the lack of Lorentz invariance in the system, that we have $\tilde{G}(t_1 - t_2, x_1 - x_2) = g_1(t_1 - t_2) g_2(x_1 - x_2)$. The Ward identity, corresponding to the first non-trivial supertranslation i.e. the Carroll boost reads,

$$Q_{f=1} \tilde{G}(t_1 - t_2, x_1 - x_2) = 0, \qquad (19)$$

which yields further:

$$\left( x_1 \partial_{t_1} + x_2 \partial_{t_2} \right) g_1(t_1 - t_2) g_2(x_1 - x_2) = (x_1 - x_2) g_2(x_1 - x_2) \dot{g}_1(t_1 - t_2) = 0,$$
$$\text{implying} \quad x g_2(x) = 0. \qquad (20)$$

This leaves us with the ultra-local solution $g_2(x) = \delta(x)$, as is apparent in the discrete version (17). However, the time dependence in (17) (or equivalently in $g_1(t_1 - t_2)$ in continuum) requires the exact nature of dynamics and is not fixed just by symmetries alone. For the discrete system, we used the solutions $\alpha_j(t) = e^{-2i\tau t} \alpha_j(0), \beta_j(t) = e^{2i\tau t} \beta_j(0)$ for the Heisenberg equations of motion of the Hamiltonian (16), and further used the relations (15) for (18). For constraining the higher point functions in this theory, higher supertranslation charges can be used.

Quantum entanglement is yet another tool to probe into the physics of many-body Hamiltonians. Entanglement entropy measures the non-local correlation between the two subsystems. The von Neumann entropy for the bipartite system is given by $S_A = \text{Tr}[\rho_A \ln \rho_A]$, where $\rho_A$ is the reduced density matrix of the subsystem $A$. For non-interacting fermions, we can obtain the eigenvalues of the reduced density matrix by the method introduced by Peschel [73,74]. For a particular choice of subsystem, one truncates the correlation matrix by only involving the local sites included in it. Then, the entanglement entropy is given by

$$S_A = -\sum_i \left[ \zeta_i \ln \zeta_i + (1 - \zeta_i) \ln (1 - \zeta_i) \right], \qquad (21)$$

where $\zeta_i$ are the eigenvalues of the truncated correlation matrix. In the present system, the required entries in the correlation matrix are given explicitly by (18) restricting to equal time conditions. Of course, there are multiple ways to choose a subsystem. We consider two distinct ways to make entanglement cuts, which divide the ladder into two halves. For the kind of cut shown in the upper panel of Fig. (2b), in the open boundary condition, the eigenvalues of the

---

[7]Note that in ultra-local quantum field theories with a holographic dual, the ground state spontaneously breaks the infinite number of supertranslation symmetries, keeping only a finite number of them. The Carroll boost, which in our notation corresponds to $Q_{f=1}$, is always intact though.

subsystem correlation matrix are all 0 and 1 (which do not contribute to the $S_A$), save for only two, which have a value of 1/2. For the other cut, as shown in the lower panel of Fig. (2b), there are three non-trivial eigenvalues $(2 \pm \sqrt{2})/4$ and 1/2. Apart from them, the rest are 0 and 1.

Focusing on the subsystem $A$ defined via the cut as per Fig. (2b), the $S_A$ is subsystem size independent and is equal to $2\ln 2$ and $\ln 2$ for periodic and open boundary conditions, respectively. Notice here that the equal time correlation matrix is independent of $\tau$, i.e. the gap. Hence, even in the gapless limit, the feature of the subsystem $S_A$ remaining agnostic to the subsystem's size remains robust, and the well-known logarithmic violations of area-law for critical gapless systems do not pop up. This could be thought of as typical feature of a *scar* state [75]. The EE spectrum can be physically understood by noticing that scar eigenstates usually can be represented as Matrix Product States (MPS) [76], and the area-law ground states of gapped systems in one dimension can be written in MPS representations with system size independent bond dimension [77].

Let us now again take a digression to understand the above results. The subsystem size independence of entanglement entropy is consistent with earlier computations in conformal Carrollian field theory in 2D (CCFT$_2$) and holographic realisations thereof. The symmetry group including conformal transformation, is generated by the BMS$_3$ algebra [29, 78]:

$$
\begin{aligned}
[L_n, L_m] &= (n-m)L_{m+n} + \frac{c_L}{12}\left(n^3 - n\right)\delta_{m+n,0}\,, \\
[L_n, M_m] &= (n-m)M_{m+n} + \frac{c_M}{12}\left(n^3 - n\right)\delta_{m+n,0}\,, \\
[M_m, M_n] &= 0\,.
\end{aligned}
\tag{22}
$$

The corresponding Lie group is the group of asymptotic (large) diffeomorphisms of theories of gravity in asymptotically flat space-times in three dimensions (see Appendix.(A)), and $c_L, c_M$ are the two central charges. A natural proposal of holography for this gravity set-up puts forward 2D conformal Carrollian theories in the asymptotic boundary. A major check for the validity of such a proposal is provided by the sub-region entanglement entropy in the boundary field theory. Let's consider a 1D subregion in the boundary having $l_x$ and $l_t$ as the spatial and temporal extents respectively. From symmetry considerations, as well as from calculations from the gravity side provide the following formula for the universal part of sub-region entanglement entropy:

$$
S = \frac{c_L}{6}\ln\left(\frac{l_x}{a}\right) + \frac{c_M}{6}\left(\frac{l_t}{l_x}\right)\,.
\tag{23}
$$

If we consider the bulk gravity theory to be described by Einstein gravity (as one option), then the central charges of the BMS$_3$ algebra take the values $c_L = 0$ and $c_M = \frac{3}{G}$. Here, $G$ is the Newton's constant. If we consider an equal time subregion, $l_t = 0$ and hence the universal part of the subregion entanglement entropy vanishes. Compared to this, the specific subsystem-size independent value $2\ln 2$ that we got from (21) is the particular model dependent one, and is even valid at the conformal point, $\tau \to 0$.

In fact, the non-zero value of entanglement entropy $2\ln 2$ is a boundary contribution that doesn't have any bearing on local physics. The zero-modes localized at the edges do not exhibit any long-range entanglement, as the edge entanglement ($S_{edge}$) (which extracts the entanglement between the two edges) vanishes irrespective of the value of the gap $\tau$.

$$
S_{\text{edge}} = S_{\text{obc}} - S_{\text{pbc}}/2 = 0\,.
\tag{24}
$$

We keep in mind that the merger of the flat bands and, hence, closing and reopening of the gap happens as one varies the parameter $\tau$ continuously from a positive to a negative value, which

is a topological phase transition [79], captured by a change in winding number.[8] This reflects the fact that edge entanglement is not always a fool-proof order parameter for topological phase transition. Similar sub-system size independence of entanglement entropy in strictly ultra-local Carrollian fermionic [80], Kitaev flat bands [81] and bosonic theories [57] have also been reported in recent literature.

## 3.2 Carroll boost and absence of Aharonov-Bohm effect

As we discussed earlier, there is no left-right chiral symmetry in a gapped dispersive free fermionic theory, as the fluctuations around the ground state move with speed less than the Fermi velocity. Any inherently boosted frame, therefore, can distinguish the left and right moving modes. In a gapless theory, though, these degrees of freedom move at Fermi velocity, which is Lorentz boost invariant.

Without going to a boosted frame, a way to see this absence of chiral symmetry would be the Aharonov-Bohm set-up. To illustrate this, we will consider such a geometry of 1D chain with imposed periodic boundary conditions. Let the chain's plane (we choose it to be a circle lying on a plane) be pierced by a magnetic field. This effect of a non-zero magnetic field is clearly manifested in a number of observables, including the gap and subsystem entanglement entropy [82]. It would be interesting to see the effect of boost and magnetic flux in the gapped flat-band system [45, 83].

We actually can go to any arbitrary supertranslated frame, a special case of which, as discussed earlier, is the Carroll boost. To see the effect of supertranslation on the dynamical features of lattice fermions, we consider the finite version of (7), under which the CLS modes transform as follows:

$$\alpha_m \to e^{iQ_f}\alpha_m e^{-iQ_f} = \alpha_m^f = e^{-2i\tau f_m}\alpha_m, \quad \text{similarly,} \quad \beta_m \to \beta_m^f = e^{2i\tau f_m}\beta_m, \tag{25}$$

where the $Q_f$ is the supertranslation generating charge (5). It is then clear that in any arbitrarily supertranslated frame, the ultra-local (CLS like) modes only receive position-dependent phases, which are just local gauge transformations. Irrespective of the choice of the lattice functions $f$, i.e. for any chosen supertranslated frame, the correlation functions would then remain completely invariant. This fact has a powerful consequence on the effect of a magnetic field on the system in an Aharonov-Bohm set-up.

To illustrate this, we will consider a fermionic chain of $\alpha$ oscillators with imposed periodic boundary conditions. We again consider the (circular) plane of the chain be pierced by a magnetic field with total flux $\Phi$. The Hamiltonian in the presence of the flux can be realized by giving a position-dependent phase to the oscillators:

$$\alpha_m \to e^{im\phi}\alpha_m, \tag{26}$$

where $\phi = \Phi/N$, $N$ being the number of oscillators in the chain. For a single flat band chain with band energy $\tau$, this can be compared with the supertranslation transformation in (25) for lattice function $f_m = -m\phi/(2\tau)$. As one can realise, this exactly corresponds to the Carroll boost transformation, i.e. the transformation under the charge $Q_{f=1}$. Physical observables in ultra-local theories, i.e. correlation functions, are independent of any supertranslation, as seen above. This predicts that consequently the flat band fermionic system is stable under Aharonov-Bohm flux insertion.

This can be intuitively traced back to the fact that in the continuum picture, the magnetic field enters the dynamics by replacing the spatial derivatives in the action/ Hamiltonian with a gauge covariant derivative:

$$\nabla \to \nabla + ieA. \tag{27}$$

---

[8]See the discussion below (12).

The flat-band system, or equivalently a continuum electric Carroll theory, is devoid of spatial derivatives; hence, the gauge/ magnetic field doesn't have any way to talk with the fermions, resulting in the invariance of physical observables under Aharonov-Bohm flux.

### 3.3 Perturbative breaking of Carroll symmetry and return probability

As of now, we have been discussing perfectly supertranslation invariant free theories. The ultra-locality as dictated by the emergent Carroll symmetry induces certain peculiar effects in the quantum structures, as we have elucidated in past sections. In the following, we will work towards the return probability of localized states when quenched by an explicitly Carroll-breaking Hamiltonian.

We start with a single particle localized state. If we quench this state $|\Psi\rangle$ with the Carroll invariant Hamiltonian, i.e. (12) with $t_1 = t_2$, the return probability $|\langle\Psi, 0|\Psi, t\rangle|^2$ will oscillate between 0 and 1, as one could expect. This is because the state $|\Psi\rangle$ is a linear combination of 4 compact localized eigenstates (15) of the Hamiltonian. However, the Hamiltonian (12), with $t_1 \neq t_2$ breaks manifest Carroll invariance and ultra-locality, and hence we could use that to quench the ultra-local Hamiltonian. This post quench Hamiltonian, written in terms of the CLS is:

$$H = 2\tau \sum_j \left(\alpha_j^\dagger \alpha_j - \beta_j^\dagger \beta_j\right) + 2\Delta \sum_j \left(\alpha_{j+1}^\dagger \alpha_{j-1} - \beta_{j+1}^\dagger \beta_{j-1} + \beta_{j+1}^\dagger \alpha_{j-1} - \alpha_{j+1}^\dagger \beta_{j-1} + \text{h.c.}\right). \quad (28)$$

Here, $t_1 = \tau + \Delta$, $t_2 = \tau - \Delta$. If we quench $|\Psi\rangle$ with this, the state will spread throughout the lattice, and the return probability/ Loschmidt echo will decay with time. Since the state $|\Psi\rangle$ is a single particle one, the evolution of this won't be affected by the interaction terms containing a product of 4 or more fermions.

Following this as a quench protocol, we set $\Delta/\tau \ll 1$ to probe the effect of Carroll breaking deformations perturbatively. To do so, we write down the Hamiltonian in Fourier space:

$$H = \sum_k \psi_k^\dagger \mathcal{H}_k \psi_k, \quad (29)$$

where the two component spinor $\psi_k^\dagger = (c_k^\dagger \ d_k^\dagger)$ and the kernel is given by $\mathcal{H}_k = \boldsymbol{a}(k) \cdot \boldsymbol{\sigma}$. The vector here is $\boldsymbol{a}(k) = (0, 2(\tau - \Delta)\sin k, 2(\tau + \Delta)\cos k)$. Using the above variables, a single site-local state $|\Psi\rangle = c_m^\dagger|0\rangle$ can be expanded in the eigenbasis of the full Hamiltonian $H$:

$$|\Psi, 0\rangle = \frac{1}{2\pi} \int_{-\pi}^{\pi} dk \frac{e^{ikm}}{N(k)} \left[(a_3 + |\boldsymbol{a}|)\tilde{c}_k^\dagger + ia_2 \tilde{d}_k^\dagger\right]|0\rangle, \quad (30)$$

with the normalization $N(k) = \sqrt{2|\boldsymbol{a}|(a_3 + |\boldsymbol{a}|)}$. Here $\tilde{c}_k^\dagger|0\rangle$ and $\tilde{d}_k^\dagger|0\rangle$ are the single particle eigenstates of $H$, respectively with eigenvalues $\pm|\boldsymbol{a}|$. The return amplitude, therefore, can be easily expressed as the overlap of the states:

$$\langle\Psi, 0|\Psi, t\rangle = \frac{1}{2\pi} \int_{-\pi}^{\pi} \frac{dk}{N^2(k)} \left[(a_3 + |\boldsymbol{a}|)^2 e^{i|\boldsymbol{a}|t} + a_2^2 e^{-i|\boldsymbol{a}|t}\right]. \quad (31)$$

Expanding the above in a power series, we see that the linear power terms in $\frac{\Delta}{\tau}$ appearing in the pre-factors of the phases do not survive the above integral. The Carroll breaking parameter $\Delta$ only enters through the phases:

$$\langle\Psi, 0|\Psi, t\rangle = \frac{1}{2\pi} \int_{-\pi}^{\pi} \cos\left(2(\tau + \Delta\cos(2k))t\right) dk = \cos(2\tau t) J_0(2|\Delta t|). \quad (32)$$

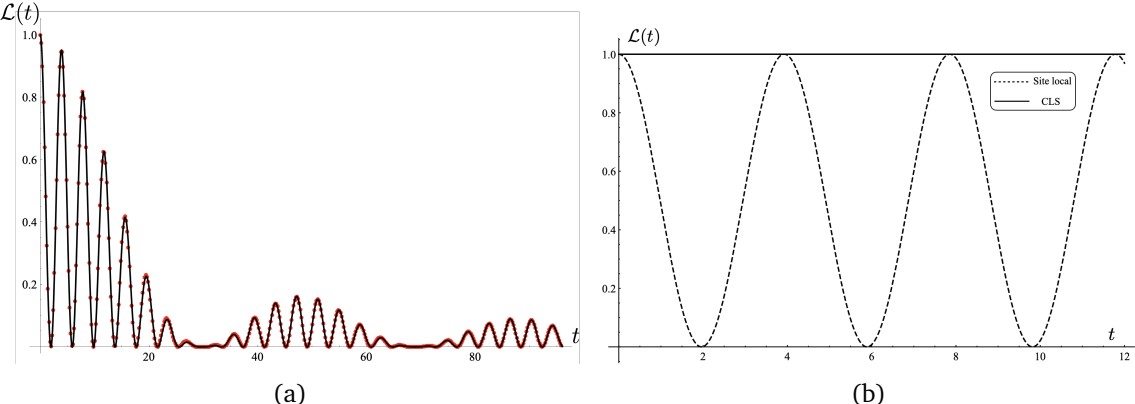

Figure 3: (a) Loschmidt echo for Carroll breaking deformation with $\tau = 0.4, \Delta = 0.04$. The solid line is the function analytically evaluated up to linear order in $\frac{\Delta}{\tau}$. The red dots are for the exact numerical evaluation of the integral without making an approximation. (b) As a comparison, the return probability is shown in the pristine Carroll invariant evolution, i.e. with $\Delta = 0$, both when the initial state is a CLS and when it is a site-local state. One can see there is no decay in the former case, manifesting the ultra-local nature.

The Loschmidt echo hence is: $\mathcal{L}(t) = \cos^2(2\tau t)J_0(2|\Delta t|)^2$, which we plot in Figure (3a).

We end this section with a comparative note about the return probability computation for the pristine Carroll symmetric theory. For example, if one starts with the initial state $|\Psi\rangle = c_m^\dagger|0\rangle$, i.e. a site local state, one gets a periodic return probability $|\langle\Psi, 0|\Psi, t\rangle|^2 = \cos^2(2\tau t)$. In other words, this means that the state remains localized within a 4-site plaquette composed of $c_m, d_m, c_{m+1}, d_{m+1}$. As expected, since the CLS are the stationary state, if one uses $|\Psi\rangle = \alpha_m^\dagger|0\rangle$, the return probability is pristine and is forever equal to 1 as in Figure (3b).

# 4 Adding interactions with supertranslation invariance

We have so far been focussed on the free theory (and Carroll-breaking deformations thereof) in the last few sections. Once we know the construction of CLS, constrained through super-translation symmetry, we can use them as building blocks for interactions, which would, by definition, be manifestly supertranslation invariant. Taking a cue from (9), (16), and then imposing symmetry constraints, we see that the only such choice involving four spinless fermions is:

$$\sim \sum_j \alpha_j^\dagger \alpha_j \beta_j^\dagger \beta_j,$$

where $\alpha$, $\beta$ oscillators are CLS made out from site-local modes (15). For the interacting theory, we will focus on the system with periodic boundary conditions in the site-local oscillators so as to make the identifications: $c_{N+1} \equiv c_1, d_{N+1} \equiv d_1$. With this set up, the total Hamiltonian is

$$H = \sum_j \left( V n_j^\alpha n_j^\beta + 2\tau \left( n_j^\alpha - n_j^\beta \right) \right). \tag{33}$$

We have used number operators $n_j^\alpha = \alpha_j^\dagger \alpha_j$ etc., and $V$ is the strength of interaction. The solutions to the Heisenberg equations of motion for the interacting system are:

$$\alpha_j(t) = e^{-it\left(2\tau + V n_j^\beta\right)}\alpha_j(0), \qquad \beta_j(t) = e^{it\left(2\tau - V n_j^\alpha\right)}\beta_j(0). \tag{34}$$

This is an exactly solvable model for all values of $V$ and $\tau$, as we can compute all the eigenvalues and eigenstates analytically. The solvability of this model is a consequence of an infinite number of global symmetries, i.e. the supertranslations (7) that persist even after adding the interaction. The corresponding conserved charges are just the modified version of (5). Explicitly:

$$Q_f = \sum_j f_j \left( V n_j^\alpha n_j^\beta + 2\tau \left( n_j^\alpha - n_j^\beta \right) \right).$$
(35)

Although not fragmented when written in the basis of site-local particle states, the Hilbert space in terms of the CLS particle states can be thought to be in a situation akin to being completely fragmented. In such a fragmented case, the total Hilbert space is split into infinitely many dynamically disconnected sectors, so that a large part of it is inaccessible to set of initial states. Our situation is, however, a very particular case of this, where every isolated block contains just one state, made out of tensor product of the CLSs. Looking at the total dimension of the Hilbert space, there are $2^N$ such states, that can only be understood from the CLS basis. These kind of fragmented Hilbert spaces are another telltale sign of scar-like states appearing in the theory,[9] as it directly relates to violation of ETH [86]. This ergodicity breaking also becomes apparent as the autocorrelation functions in the CLS basis keep oscillating to finite values over long times. We would come back to the details these structures in a separate communication.

One may compare the theory (33) with the local Hubbard interaction introduced on the otherwise flat-band free theory in [45]. Such an interaction term in terms of our notation (12) would be $\sum_j c_j^\dagger c_j d_j^\dagger d_j$. As one can check, this is not a supertranslation invariant interaction term, breaking the ultra-locality. So, the manifest supertranslation invariance is confined to the Hamiltonian written in the CLS basis states, making the Carroll symmetry manifest in this basis.

# 5 Quantum Phases: The half-filling case

Seemingly trivial because of exact solvability, the theory (33) exhibits a number of non-trivial quantum phases. In the free theory (16), if we tune $\tau$ from a positive to a negative value, one can see that the ground state switches from the one having all $\beta$ oscillators excited to the one having all the $\alpha$ ones excited. In both of these phases, the ground state is half-filled. For simplicity, we will call these two phases the Vanilla$_\alpha$ and Vanilla$_\beta$ phases, since they are easier to deal with. It is interesting to map all the emergent quantum phases either (i) by keeping the half-filling constraint or (ii) by keeping the particle filling unconstrained. In the current section, we will focus on the first of these cases.

Continuing the discussion in (3.3), we will further study the effect of Carroll breaking term ($t_1 \neq t_2$) in all the emerging quantum phases. In addition to the perturbation corrections, we flourish some DMRG numerical results[10] beyond the perturbative regime using matrix product states Ansatz as implemented in [88]. The ladder system, as shown in Fig (2)(a), is mapped into the one-dimensional chain. We then compute the ground state energy imposing open as well as periodic boundary conditions with the convergence of the order $10^{-8}$. The quantity of interest is the correlation matrix, which manifestly shows the deformations of the flat band.

---

[9]In fact the appearance of scar states via construction of CLSs as eigenstates of flat band systems has been described before, see for example [84, 85]. These CLSs remain exact eigenstates even in the presence of density-density interactions.

[10]See, for example, the classic introduction [87].

## 5.1 Phase structure

Keeping $\tau > 0$, if we turn on a small positive value of $V$, we see directly from (33), that the ground state energy is $-2\tau N$ and the $N$-fold degenerate first excited subspace has energy $4\tau - 2\tau N$. Hence, the gap is $4\tau$, which has the same value as in free theory, as expected from a two flat-band system with band energies $\pm 2\tau$. Schematically, we can denote this ground state and a generic first excited state as:

$$\left| \begin{array}{c} \circ\circ\ldots\circ\circ \\ \bullet\bullet\ldots\bullet\bullet \end{array} \right\rangle, \qquad \left| \begin{array}{c} \circ\circ\ldots\circ\overset{n}{\overbrace{\bullet}}\circ\ldots\circ \\ \bullet\bullet\cdots\bullet\underset{n}{\underbrace{\circ}}\bullet\cdots\bullet \end{array} \right\rangle. \tag{36}$$

Here, we denote the upper (lower) row as those of $\alpha$ ($\beta$) occupations, and a filled circle means an occupation of 1, whereas an ordinary circle means zero occupation.

On the other hand, for a small negative value of $V$ in (33), the first excited subspace has energy $V + 4\tau - 2N\tau$, now taking the gap to $V + 4\tau$. A basis for the first excited subspace consists of $N(N-1)$ vectors, which can be expressed schematically as:

$$\left| \begin{array}{c} \circ\circ\ldots\circ\overset{m}{\overbrace{\bullet}}\circ\ldots\circ \\ \bullet\cdot\bullet\underset{n\neq m}{\underbrace{\circ}}\bullet\ldots\bullet\bullet \end{array} \right\rangle. \tag{37}$$

These gap values are non-perturbative. From here, we can readily infer that for $\tau > 0$, the gap vanishes for $V = -4\tau$. We must emphasize here that in the purely Carrollian theory, there is no part of the spectrum that is continuous (or quasi-continuous in the lattice sense), and each energy level is highly degenerate. Hence, even when the gap between the ground and the first excited state closes, the ground state becomes degenerate. There emerges a finite macroscopic gap above it, and the ground state is never a part of a continuous spectrum. Articulated otherwise, the density of states function is only a sum of Dirac delta functions supported at distinct points. For this reason, signatures of finite 'energy' scales may persist in correlation functions.

For this phase, i.e. $\tau > 0, V > -4\tau$, the ground state is unaffected by the value of $V$ and is the same as that for $V = 0$ (36). As a result, the ground state entanglement entropy computation remains the same as our discussion in section (3.1). Curiously, $V < -4\tau$ is a different quantum phase altogether from the vanilla ones described above, with a degenerate ground state manifold. A basis for this space consists of states for which

$$n_j^{\alpha} = n_j^{\beta}, \forall j \quad \text{and} \quad \sum_{j=1}^{N} n_j^{\alpha} = N/2 = \sum_{j=1}^{N} n_j^{\beta}. \tag{38}$$

This is an $^{N}C_{N/2}$ dimensional space, and the ground state energy is $NV/2$. Any vector of this space has zero overlaps with the all-$\beta$ filled state (36), i.e. the unique ground state of the $V > -4\tau$ phase. Hence, the fidelity[11] across the phase transition line $V = -4\tau, \tau > 0$ identically vanishes.

A very similar phase transition appears for $\tau < 0$ as well. The ground state for $\tau < 0, V > 4\tau$ has all the $\alpha$ and none of the $\beta$ oscillators excited. The gap between the ground state and the first excited one vanishes along $\tau < 0, V = 4\tau$. For $\tau < 0, V < 4\tau$, however, one reaches the phase where the lowest energy level is degenerate, and the basis for the

---

[11]Fidelity signifies the modulus of the overlap between two states, or the transition probability between one state to the other. In the purview of information theory, this denotes how close two states are to each other. For the significance of fidelity across quantum phase transitions, see [89] for example.

Table 1: Comparative characteristics of the three phases associated to half-filling case.

| Phase | Parameter space | Ground State | |
|---|---|---|---|
| Vanilla$_\alpha$ | $\tau < 0, V > 4\tau$ | $\lvert n_j^\alpha = 1, n_j^\beta = 0, j = 1, ..., N \rangle$ | |
| Vanilla$_\beta$ | $\tau > 0, V > -4\tau$ | $\lvert n_j^\alpha = 0, n_j^\beta = 1, j = 1, ..., N \rangle$ | |
| Exotic$_{\alpha\beta}$ | $V < 0, -V/4 > \tau > V/4$ | $\lvert n_j^\alpha = n_j^\beta, j = 1, ..., N \rangle,$ | $\sum_{j=1}^{N} n_j^\alpha = \frac{N}{2} = \sum_{j=1}^{N} n_j^\beta$ |

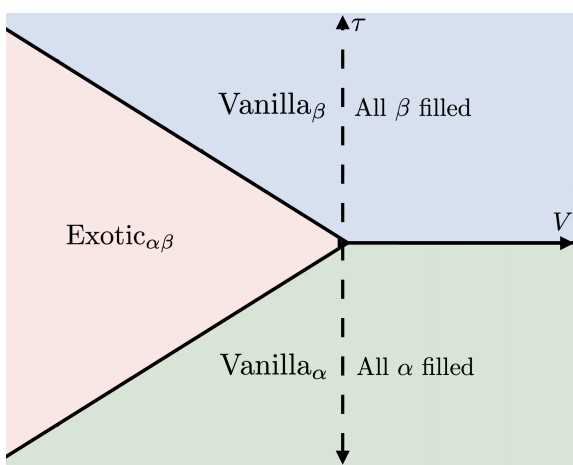

Figure 4: The $(\tau - V)$ phase plot for our interacting system showing the vanilla phases (blue and green) and the exotic phase (pink). The gap between the first exited and ground states vanishes along the thick lines; hence, they are phase boundaries. The fillings in the three different phases are shown in the Table (1). The three phase boundary lines are: (i) $\tau = 0, V > 0$, (ii) $\tau > 0, V = -4\tau$ and (iii) $\tau < 0, V = 4\tau$.

ground state subspace is the same as (38). This phase persists for $V < 0, -V/4 > \tau > V/4$, as depicted in the phase plot Fig. (4). This phase, which we will dub as Exotic$_{\alpha\beta}$, or simply the exotic phase, has a lot more intriguing physics attached to it, and that is what we will discuss next.

Using the time evolution equations (34), we find the time-dependent correlation functions in the Vanilla$_\beta$ phase. Since the ground state in this phase doesn't have any $\alpha$ excitations, the correlation functions of site-local modes remain exactly the same as those in (18) or (17). As long as one is in this phase and lowers $\tau$ from positive values to the topological phase transition at $\tau \to 0^+$, the time dependence vanishes, i.e.

$$\lim_{\tau \to 0^+} \langle \beta_i^\dagger(t)\beta_j(0) \rangle = \delta_{i,j}. \tag{39}$$

It is instructive to go to the continuum limit in this case, where we put $\beta_j \to \sqrt{a}\psi(x)$ ($a$ is the lattice constant), near the phase transition point, one gets:

$$\langle \psi^\dagger(x, t)\psi(y, 0) \rangle \sim \delta(x - y). \tag{40}$$

To match with this, we recall that for 1+1 dimensional quantum field theories with supertranslation invariance, correlation functions of spinless operators of definite scaling dimensions at a conformal point are known [36, 51, 90, 91] to be of the following form, derived purely from symmetry arguments:

$$\langle O_\Delta(x, t)O_{\Delta'}(y, 0) \rangle \sim t^{1-\Delta-\Delta'}\delta(x - y). \tag{41}$$

This expected form matches exactly with the continuum version correlation function (40), since fermions in 1+1 dimensions have scaling dimension $\Delta = 1/2$ (not to be confused with the strength of the Carroll breaking perturbation in (28).).

On the other hand, if one naively probes gaplessness with correlation functions in the Vanilla$_\beta$ phase near the line $V = -4\tau$ for finite $\tau$, we will see that the correlation functions remain of the form (18), i.e.

$$\langle \beta_i^\dagger(t)\beta_j(0)\rangle = e^{-2i\tau\,t}\delta_{i,j}, \qquad \langle \alpha_i^\dagger(t)\beta_j(0)\rangle = 0 = \langle \alpha_i^\dagger(t)\alpha_j\rangle. \tag{42}$$

Clearly, $\tau$ bears the signature of the finite gap above the degenerate ground state. This is in apparent contradiction with (41). However, one has to remember that the Ward identity, solved to get the generic structure of (41), was valid in a continuum theory. In the continuum, $\tau$ is a relevant coupling, whereas $V$ is marginal. This being an exactly solvable model, there won't be any loop correction in computing RG flows towards the IR. If one has to reach a finite $\tau$ at low energy, one has to start with $\tau \approx 0$ at the continuum. Hence, in the continuum model, (42) gives back the expected result (41) with $\Delta = 1/2 = \Delta'$.

## 5.2 Dynamical features of the exotic phase

Extracting dynamical features out of the ground state in the Exotic$_{\alpha\beta}$ phase is a challenging task because the ground state subspace of this phase is highly degenerate. To address this issue, one should make a physically motivated prescription for a unique choice of the ground state, as far as calculation of quantities like correlation functions are concerned. We will motivate such a choice towards the end of this section.

First let us try to understand how our eigenstates, written in the CLS basis, work. Evidently, none of the basis states (38) are covariant under the lattice translation operation. This actually is true for any generic ultra-local energy eigenstate of the Hamiltonian, i.e.:

$$\left(\prod_{i=1}^{N_1 \leq N} \alpha_{m_i}^\dagger\right)\left(\prod_{j=1}^{N_2 \leq N} \beta_{n_j}^\dagger\right)|0\rangle. \tag{43}$$

Here the labels $\{m_i | i = 1,\ldots,N_1 \leq N\}$ etc. are two ordered sets of positive integers with maximum value $N$. To see this, let $U$ be the generator of unit lattice translation towards the right. Then its action on the site local modes $c, d$ and consequently on the CLS basis oscillators are:

$$U \circ c_j = c_{j+1}, \quad U \circ d_j = d_{j+1} \qquad \Rightarrow \qquad U \circ \alpha_j = \alpha_{j+1}, \quad U \circ \beta_j = \beta_{j+1}. \tag{44}$$

On the other hand, the on-shell action of supertranslation on the CLS (using (35) and the Heisenberg equation of motion derived from (33)) are very different:

$$Q_f \circ \alpha_j = -i f_j \left(V \alpha_j n_j^\beta + 2\tau \alpha_j\right), \quad \text{etc.} \tag{45}$$

Clearly, the unitary actions of lattice translation $U$ and supertranslation $Q_f$ do not commute for our interacting system. Since a generic ultra-local energy eigenstate (43) is an eigenstate of all supertranslations, it is not translational covariant (i.e. not invariant up to a phase factor). This is why, for the free theory, despite being energy eigenstates, the CLS are not in general Bloch wavefunctions and don't have crystal momentum as a quantum number. The exceptions to this are the ground states of the other two (Vanilla) phases, which are all $\beta(\alpha)$-filled states and hence are translation invariant.

We take this opportunity to take a detour again, and try to understand this situation in the continuum limit. In a $d$ dimensional continuum space-time, the generators of spatial translation and Carroll supertranslations can be represented as the vector fields

$P_i = \partial_i$ and $Q_f = f(\boldsymbol{x})\partial_t$ respectively. As evident, the function $f = 1$ corresponds to time translation, and the crucial part of the BMS/Carroll algebra is

$$\left[P_i, Q_f\right] = Q_{\partial_i f}\,. \tag{46}$$

An ultra-local free massive scalar field theory [36, 57], devoid of the gradient term, and endowed with supertranslation symmetries is:

$$S = \int dt\, d^{d-1}\boldsymbol{x} \left[\frac{1}{2}(\partial_t \phi)^2 - \frac{1}{2}m^2\phi^2\right]. \tag{47}$$

The quantization of this theory is fairly simple, considering the following solution of the equation of motion:

$$\phi(t,\boldsymbol{x}) = \frac{1}{\sqrt{m}}\left[a^\dagger(\boldsymbol{x})e^{imt} + a(\boldsymbol{x})e^{-imt}\right]. \tag{48}$$

The modes $a(\boldsymbol{x})$ as quantum operators satisfy:

$$\left[a(\boldsymbol{x}), a^\dagger(\boldsymbol{x}')\right] = \frac{1}{2}\delta^{(d-1)}(\boldsymbol{x} - \boldsymbol{x}')\,. \tag{49}$$

Just as in the case of fermionic CLS we consider in this work, the bosonic modes $a$ are ultra-local operators, and the (normal ordered) Hamiltonian

$$H = 2m \int d^{d-1}\boldsymbol{x}\, a^\dagger(\boldsymbol{x})a(\boldsymbol{x})\,, \tag{50}$$

is already diagonal in this representation. This is in contrast to field theories having spatial derivative terms in the Hamiltonian, where one needs to expand the local fields in terms of Fourier transform modes for diagonalizing the Hamiltonian. The vacuum $|0\rangle$, annihilated by all the ultra-local modes $a(\boldsymbol{x}), \forall \boldsymbol{x}$ is the unique ground state. On the other hand, the single excitation states $|\boldsymbol{x}\rangle := a^\dagger(\boldsymbol{x})|0\rangle$ are of energy $m$. Now, one should notice these states don't carry a momentum quantum number, as they are not eigenstates of the momentum operator $\boldsymbol{P} = \int d^{d-1}\boldsymbol{x}\,\pi\boldsymbol{\nabla}\phi$. This is very much analogous to the lattice fermionic CLS we discussed above. The only difference is that, on the lattice, one doesn't have a continuous translation. Hence, the interesting operator is the unitary discrete translation generator $U$ (44).

Coming back to our original problem, one way to construct a translation invariant state is to pick up any local basis state $|\psi\rangle$ given by

$$n_j^\alpha = n_j^\beta\,, \quad \forall j \qquad \text{and} \qquad \sum_{j=1}^N n_j^\alpha = N/2 = \sum_{j=1}^N n_j^\beta\,, \tag{51}$$

and perform a symmetrization under the discrete Abelian translation group. The process involves acting on the state by the unit translation generator of the Abelian translation group repeatedly and finally taking a linear combination of all such generated states with equal amplitude to get:

$$|\psi\rangle_{\text{Sym}} = \frac{1}{\sqrt{M+1}}\sum_{p=0}^{M \leq N} U^p|\psi\rangle\,. \tag{52}$$

Here we assumed $U^{M+1}|\psi\rangle = |\psi\rangle$ and name $M$ the symmetrization rank for a particular local state $|\psi\rangle$. This is an eigenstate of the unit lattice translation operator $U$ with eigenvalue 1. Hence, this qualifies as a translation invariant Bloch state with crystal momentum 0 mod $2\pi/a$. Translation symmetrized states like (52), having vanishing crystal momentum, do form a subspace[12] of the ground state Hilbert space.

---

[12]We do not attempt to solve the combinatoric problem of finding the dimension of this subspace.

We will consider two extreme cases of local states from above, whose translation symmetrization ranks are maximal, i.e. $M = N$ and $M = 1$, respectively. One choice for $M = N$ is when $|\psi\rangle$ is a domain wall state, ie.:

$$|\psi_{\text{DW}}\rangle = \left| \begin{array}{c} \overbrace{\bullet\bullet\ldots\bullet\bullet}^{N/2}\circ\circ\ldots\circ\circ \\ \underbrace{\bullet\bullet\ldots\bullet\bullet}_{N/2}\circ\circ\ldots\circ\circ \end{array} \right\rangle. \tag{53}$$

Before proceeding further, we will make a curious observation regarding the effect of Carroll supertranslations on the above translation invariant 'zero' momentum states $|\psi\rangle_{\text{Sym}}$ described by (52). This stems from the on-shell transformation properties of the CLS under supertranslations (45). The finite version of this transformation on CLS oscillators is:

$$\alpha_m \rightarrow \alpha_m^f = e^{-if_m(2\tau + V n_m^\beta)}\alpha_m, \quad \text{similarly} \quad \beta_m \rightarrow \beta_m^f = e^{if_m(2\tau - V n_m^\alpha)}\beta_m. \tag{54}$$

Using these transformations, it is straightforward to observe that under an arbitrary supertranslation generated by a lattice function $f$, the domain wall state acquires an $f$ dependent phase:

$$|\psi_{\text{DW}}\rangle \rightarrow |\psi_{\text{DW}}\rangle^f = \exp\left( 2iV \sum_{j=1}^{N/2} f_j \right) |\psi_{\text{DW}}\rangle. \tag{55}$$

On the other hand, the unit-translated state acquires a different phase:

$$U|\psi_{\text{DW}}\rangle \rightarrow (U|\psi_{\text{DW}}\rangle)^f = \exp\left( 2iV \sum_{j=2}^{N/2+1} f_j \right) U|\psi_{\text{DW}}\rangle. \tag{56}$$

This clearly indicates that for a non-constant $f$, $|\psi_{\text{DW}}\rangle_{\text{Sym}}$ won't remain invariant (covariant up to a phase) under supertranslation. This conclusion is in harmony with the discussion previously presented in this section, in particular, concerning the effect of non-commutation of supertranslation and spatial translation. However, in the translation-symmetrized state, $|\psi_{\text{DW}}\rangle_{\text{Sym}}$, the correlation functions still remain manifestly ultra-local:[13]

$$\begin{aligned} \langle \alpha_i^\dagger \alpha_j \rangle = \frac{1}{2}\delta_{i,j} = \langle \beta_i^\dagger \beta_j \rangle, & \qquad \langle \alpha_i^\dagger \beta_j \rangle = 0, & \text{or} \\ \langle c_i^\dagger c_j \rangle = \frac{1}{2}\delta_{i,j} = \langle d_i^\dagger d_j \rangle, & \qquad \langle c_i^\dagger d_j \rangle = 0, & \text{in terms of site-local operators.} \end{aligned} \tag{57}$$

Similar conclusions can be shown to hold for the other choice $M = 1$, which strictly has the altered filling state:

$$|\psi_{\text{Alt}}\rangle = \left| \begin{array}{c} \bullet\circ\bullet\circ\cdots\bullet\circ\bullet\circ \\ \bullet\circ\bullet\circ\cdots\bullet\circ\bullet\circ \end{array} \right\rangle. \tag{58}$$

Correlation functions in its symmetrized version, $|\psi_{\text{Alt}}\rangle_{\text{Sym}}$ has again, persistent ultra-local feature:

$$\langle \alpha_i^\dagger \alpha_j \rangle = \frac{1}{2}\delta_{i,j} = \langle \beta_i^\dagger \beta_j \rangle, \qquad \langle \alpha_i^\dagger \beta_j \rangle = 0. \tag{59}$$

---

[13]These Kronecker deltas are periodic, for the periodic boundary conditions imposed.

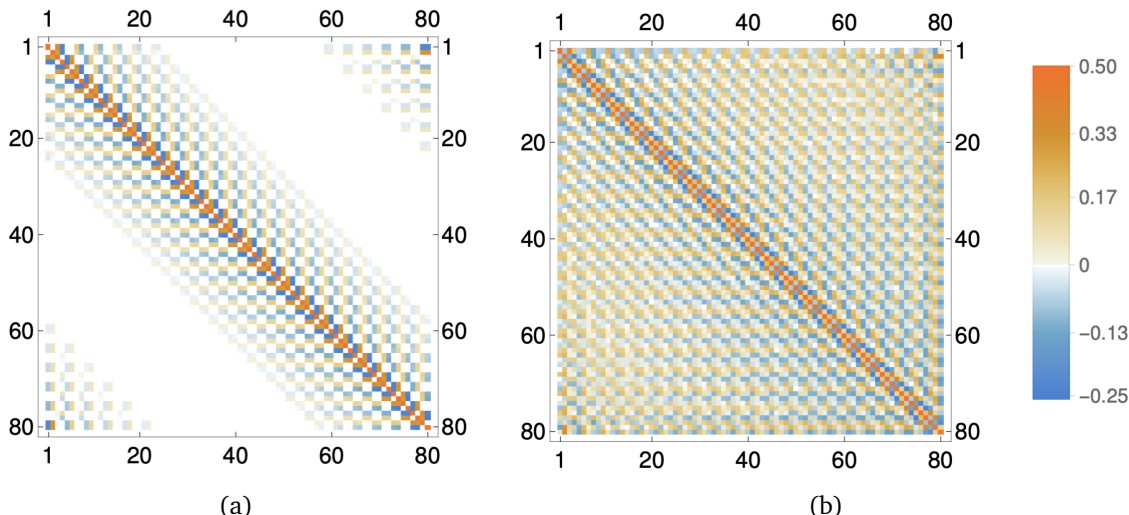

Figure 5: Spreading of ultralocal correlations in the vanilla phase under the Carroll breaking deformation. Here, we have $\Delta = 0.01, \tau = 0.4$, (a) $V = -0.4$, (b) $V = -1.6$ (This exactly is the critical point) with open boundaries.

## 5.3 Effect of Carroll breaking deformations at half-filling

In the free theory we probed the instability of the CLS/ ultra-local features of the flat band model under a small Carroll-breaking perturbation probed through the return probability, in section (3.3). That probe of course would not have been able to capture the intricacies of the three different quantum phases, since single particle states are agnostic to the interaction. In this section, we will take a more direct approach of observing the effect of Carroll breaking deformations on the spectrum and correlation functions at half-filling.

Let us consider the interacting analogue of the full free Hamiltonian (28) for $\Delta \neq 0$ and the interaction term and choose $\Delta$ to be treated perturbatively. So the total Hamiltonian can be split into ultra-local $H_{\mathrm{ul}}$ and dispersive $H_{\mathrm{d}}$ parts:

$$H = H_{\mathrm{ul}} + H_{\mathrm{d}}. \tag{60}$$

Where in the present case $H_{\mathrm{ul}}$ is the ultralocal interacting Hamiltonian (33) and $H_{\mathrm{d}}$ is the perturbation from the second line of (28).

**Unstable ultra-locality in the vanilla phases**

Let us now revisit the phase structure of $H_{\mathrm{ul}}$ and, especially for this subsection, focus at a point $0 > V > -4\tau, \tau > 0$, preferably near the critical line $V = -4\tau$ as per the figure (4). For this particular point in the parameter space, denote the ground state $|\mathrm{grnd}\rangle$ of $H_{\mathrm{ul}}$, which has all the $\beta$ modes excited and no $\alpha$, as:

$$|\mathrm{grnd}\rangle = \left| \begin{array}{c} \circ\circ\ldots\circ\circ \\ \bullet\bullet\ldots\bullet\bullet \end{array} \right\rangle, \qquad H_{\mathrm{ul}}|\mathrm{grnd}\rangle = \varepsilon_0|\mathrm{grnd}\rangle, \qquad \varepsilon_0 = -2N\tau. \tag{61}$$

The upper row is for $\alpha$ modes, and the lower with illed-in disks is for the $\beta$ ones. In the influence of the perturbation $H_{\mathrm{d}}$, it is easy to check that this nondegenerate ground state energy doesn't receive any first-order corrections. However, the ground state itself is corrected

in the first order of $\Delta$ as the perturbation term rearranges filled-in sites:

$$|\text{grnd}_\Delta\rangle = |\text{grnd}\rangle + \frac{\Delta}{2\tau}\sum_j\left(\left|\begin{array}{c}\overbrace{\phantom{\circ\circ\ldots\circ\,\bullet\,\circ\ldots\circ}}^{j+1}\\ \circ\circ\ldots\circ\,\bullet\,\circ\ldots\circ\\ \bullet\ldots\bullet\underbrace{\,\circ\,}_{j-1}\bullet\ldots\bullet\bullet\end{array}\right\rangle - \left|\begin{array}{c}\overbrace{\phantom{\circ\ldots\circ\,\bullet\,\circ\ldots\circ\circ}}^{j-1}\\ \circ\ldots\circ\,\bullet\,\circ\ldots\circ\circ\\ \bullet\bullet\ldots\bullet\underbrace{\,\circ\,}_{j+1}\bullet\ldots\bullet\end{array}\right\rangle\right). \quad (62)$$

This immediately results in a spreading in correlations beyond ultra-locality, in comparison to (17):

$$\begin{aligned}\langle\text{grnd}_\Delta|\alpha_m^\dagger\alpha_n|\text{grnd}_\Delta\rangle &= 0\,,\\ \langle\text{grnd}_\Delta|\beta_m^\dagger\beta_n|\text{grnd}_\Delta\rangle &= \delta_{m,n}\,,\\ \langle\text{grnd}_\Delta|\alpha_m^\dagger\beta_n|\text{grnd}_\Delta\rangle &= \left(\frac{\Delta}{2\tau}\right)\left(\delta_{m,n+2}-\delta_{m,n-2}\right).\end{aligned} \quad (63)$$

The perturbed correlation functions up of the cite-local operators are:

$$\langle\text{grnd}_\Delta|c_m^\dagger c_n|\text{grnd}_\Delta\rangle = \langle\text{grnd}|c_m^\dagger c_n|\text{grnd}\rangle + \frac{\Delta}{4\tau}\left(\delta_{m,n+1}+\delta_{m,n-1}-\delta_{m,n-3}-\delta_{m,n+3}\right),$$

$$\langle\text{grnd}_\Delta|c_m^\dagger d_n|\text{grnd}_\Delta\rangle = \langle\text{grnd}|c_m^\dagger d_n|\text{grnd}\rangle + \frac{\Delta}{4\tau}\left(\delta_{m,n-1}-\delta_{m,n+1}+\delta_{m,n-3}-\delta_{m,n+3}\right), \quad (64)$$

$$\langle\text{grnd}_\Delta|d_m^\dagger d_n|\text{grnd}_\Delta\rangle = \langle\text{grnd}|d_m^\dagger d_n|\text{grnd}\rangle + \frac{\Delta}{4\tau}\left(\delta_{m,n-3}+\delta_{m,n+3}-\delta_{m,n+1}-\delta_{m,n-1}\right).$$

The equation (64) above clearly indicates the spreading of correlation beyond the CLS plaquettes, at first order in perturbation and hence the emergence of the finite non-zero correlation length. This has been numerically expressed in terms of the correlation matrix using DMRG for particular values of $\Delta$, $\tau$ and $V$, in Fig. (5a).

As one nears the critical line from the Vanilla$_\alpha$ phase and perturbation parameter $\Delta$ becomes the same as the order of the gap $4\tau + V$, perturbative expansion is no longer reliable. To probe the correlation structure for this case, one has to resort to numerics. We employed tensor network-based DMRG calculations, as shown in the correlation matrix plot in Fig. (5b). In this figure, at the point where one reaches the exact critical line in the ultra-local theory, i.e. $V = 4\tau = -1.6$, turning on Carroll-breaking deformation with a strength $\Delta = 0.01$, completely destroys ultra-locality with the emergence of infinite (comparable to system size) correlation length. One can repeat the same exercise for the Vanilla$_\beta$ phase as well.

**Effect of $\Delta$ on gap in the vanilla phases**

Note that, the Vanilla$_\beta$ phase ground state and an element of the local basis for the first excited subspace are given by (36). Let us denote this first excited state as

$$|\phi_n\rangle = \left|\begin{array}{c}\overbrace{\phantom{\circ\circ\ldots\circ\,\bullet\,\circ\ldots\circ}}^{n}\\ \circ\circ\ldots\circ\,\bullet\,\circ\ldots\circ\\ \bullet\bullet\ldots\bullet\underbrace{\,\circ\,}_{n}\bullet\ldots\bullet\end{array}\right\rangle. \quad (65)$$

It is easy to see that the matrix elements of the dispersive perturbation $H_\text{d}$ in this subspace vanish identically:

$$\langle\phi_m|H_\text{d}|\phi_n\rangle = 0\,. \quad (66)$$

This implies that the degeneracy of the first excited state space is not lifted. As we have already seen, the ground state energy to first order in $\Delta$ is unchanged. Hence, the gap is intact up to first-order corrections in $\Delta$.

**Robust ultra-locality in the exotic phase**

Of course, one would now want to ask what happens for the Exotic$_{\alpha\beta}$ phase. The ground state subspace in this phase is spanned by the ${}^N C_{N/2}$ states (51). However, the matrix elements of $H_d$ also vanish in this subspace. Consequently, $H_d$ fails to break the degeneracy in this subspace. As discussed previously in section (5.2), one can arrange the basis for this subspace either in terms of eigenstates of supertranslation generators or that of lattice momentum. Correlations in the former ones are automatically ultra-local, and hence, due to the robustness under $H_d$, remain so. On the other hand, ultra-locality in correlations in the zero-momentum symmetrized states exemplified in (57) and (59) are robust in this perturbation too.

## 5.4 Thermodynamics

We will now focus on the thermodynamic properties of our interacting model. Partition functions for ultra-local Carrollian theories, either from the point of view of quantum mechanical path integral or from the statistical mechanical one, require a degree of subtlety as explained in [58]. One of the obvious ways is to write the partition function of the relativistic theory and take the limit of the speed of light/ Fermi velocity to zero. However, this procedure renders UV and IR divergent answers that can't be regulated for both cases of a free massive scalar [92] and a compact massless scalar. This creates an apparent problem in discussing the thermodynamics of such systems.

One way to view this subtlety, as explained in [58] is by reviewing how a functional determinant in a field theory with spatial gradient is calculated. A standard path integral computation doesn't sum over discontinuous field configurations. However, if one has an ultra-local theory without neighbourhood couplings on a lattice (for example the Einstein model of lattice vibrations), there's no a priori reason to preclude randomly discontinuous field configurations. Hence, the former route of taking characteristic speed to zero in the path integral of a local theory with a gradient term will not give the same answer as that of the continuum limit of an ultra-local theory on a lattice [57].

A definite way to extract physically meaningful partition functions is to start from a lattice-regularized version. This is most natural for Carrollian theory because of the absence of neighbourhood couplings/ gradient terms in action. In this set-up, especially, one naturally avoids plane wave expansion, and all possible discontinuous field configurations do contribute to the partition function, making it finite. In the context of our present discussion, particularly the free theory with manifest flat bands scenario, we chose CLS over Bloch states as the basis of quantization. In light of the above discussion from a thermodynamic perspective, and as explained in greater detail in [58], the preference given to CLS states is once more substantiated over Bloch states.

Let's now take a particular example at hand. For a massless compact Carroll scalar field on a 1-dimensional spatial lattice, the canonical partition function takes the following form:

$$Z = Z_{\text{site}}^N, \quad \text{where} \quad Z_{\text{site}} = \sum_{n \in \mathbb{Z}} e^{-\beta n^2/(2R^2)}. \tag{67}$$

Here, $R$ is a length scale originating from the lattice spacing and the compactification radius for the scalar. In a relativistic theory, this radius would be responsible for exhibiting T-duality. The Carroll case (67) corresponds to decoupled rigid rotors or $Z_{\text{site}}$ is the single particle partition function of a particle in an infinite well. Even though the equivalent relativistic theory is gapless, the Carrollian partition function shows up with an explicit gap. On the other hand, for a non-compact massive Carroll scalar (of mass $m$), the partition function takes the form:

$$Z = Z_{\text{site}}^N, \quad \text{where} \quad Z_{\text{site}} = \sum_{n=0}^{\infty} e^{-n m \beta}. \tag{68}$$

The examples (67) and (68) are both for free theories, and they show a gapped behaviour, ubiquitous to statistical mechanics of gapped quantum mechanical particles on a lattice. In the following, we will see a clear signature of the gapless phase for the interacting fermionic theory we considered earlier (33).

Here, we explore the thermodynamic properties of the interacting theory discussed before at half-filling. It makes sense to consider a canonical ensemble when the particle number is fixed, but to avoid complex combinatorics involved in such a description of constrained sums, we resort to a grand canonical ensemble. For the Hamiltonian (33), the grand canonical ensemble partition function is

$$Q(\beta, z) = \left(1 + 2z \cosh 2\tau\beta + z^2 e^{-V\beta}\right)^N, \tag{69}$$

where $z = e^{\mu\beta}$ is the fugacity. Given the context, we hope the inverse temperature $\beta$ may not be confused with the fermionic oscillators denoted by the same notation. As maintained throughout the paper, $N$ is the number of sites on a single ladder rung. The ensemble average particle number per site:

$$\langle n \rangle_\beta = \frac{1}{2N} z \partial_z \ln Q = \frac{z \cosh(2\tau\beta) + z^2 V e^{-\beta V}}{1 + 2z \cosh(2\tau\beta) + z^2 e^{-\beta V}}. \tag{70}$$

On the other hand, thermal average energy density,

$$\langle \epsilon \rangle_\beta = -\frac{1}{2N} z \partial_\beta \ln Q = -\frac{2z\tau \sinh(2\tau\beta) - z^2 V e^{-\beta V}/2}{1 + 2z \cosh(2\tau\beta) + z^2 e^{-\beta V}}. \tag{71}$$

At this point, forcing in the constraint of half-filled lattice, ie. $\langle n \rangle_\beta = 1/2$ in (70), and eliminating the fugacity, we can evaluate the energy density at low temperatures $T \ll \tau$, which becomes:

$$\langle \epsilon \rangle_\beta \approx -\frac{\tau}{2} \frac{sx^2 + 2x^{s/2}}{x^{s/2} + 2x^2}, \qquad \text{where} \qquad x = e^{-\tau\beta}, \quad s = -V/\tau. \tag{72}$$

At low temperatures, ie. for $x \ll 1$,

$$\langle \epsilon \rangle_\beta = \begin{cases} -s\tau/4 + \frac{\tau}{8}(s-4)x^{s/2-2}, & s > 4, \\ -\tau + \frac{\tau}{2}(4-s)x^{2-s/2}, & s < 4. \end{cases} \tag{73}$$

This clearly captures the criticality at $s = 4$, i.e. $V = -4\tau$, and also the vanishing of the gap at that point, exhibited through the vanishing of the exponent of $x$. This very satisfactorily matches up with our discussions in this section.

## 6 Quantum phases: Without filling constraint

From the perspective of a fermionic many-body system, although it is imperative that the number of particles should be fixed in the absence of a gauge field, just from quantum mechanics of the Hamiltonian alone, it may still be instructive to understand the spectrum of the theory after switching off the chemical potential. This makes the filling factor unconstrained. Consequently, the quantum phase structure drastically differs from the half-filled one we discussed in the last section.

### 6.1 Phase structure

Just as in the half-filled case, with $V > 0$, the ground state of the interacting Hamiltonian (33) is still half-filled irrespective of even without any constraints on the filling factor. Hence the

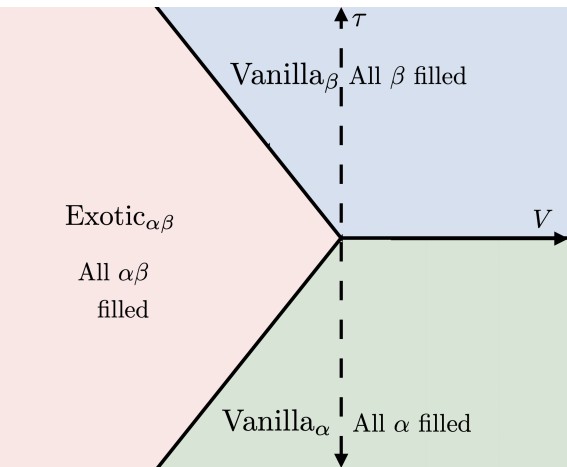

Figure 6: The $(\tau - V)$ phase plot for the interacting theory without filling constraint. The vanilla phases (blue and green) and the exotic phase (pink) have been shown. The gap between the first exited and ground states vanishes along the thick lines; hence, they are phase boundaries. The three phase-separation lines are: (i) $\tau = 0, V > 0$, (ii) $\tau > 0, V = -2\tau$ and (iii) $\tau < 0, V = 2\tau$.

Table 2: Comparative characteristics of the three phases associated with zero chemical potential.

| Phase | Parameter space | Ground State |
|---|---|---|
| Vanilla$_\alpha$ | $\tau < 0, V > 2\tau$ | $|n_j^\alpha = 1, n_j^\beta = 0, j = 1, ..., N\rangle$ |
| Vanilla$_\beta$ | $\tau > 0, V > -2\tau$ | $|n_j^\alpha = 0, n_j^\beta = 1, j = 1, ..., N\rangle$ |
| Exotic$_{\alpha\beta}$ | $V < 0, -V/2 > \tau > V/2$ | $|n_j^\alpha = 1, = n_j^\beta = 1, j = 1, ..., N\rangle$ |

Vanilla$_\alpha$ and Vanilla$_\beta$ phases still persist for $V > 0$. The same ground state structures are still there with a non-vanishing gap as one lowers $V$ below zero. However, for small (compared to $\tau > 0$) and negative $V$ the first excited state is not half-filled. For the Vanilla$_\beta$ phase, the first excited state space has states with all the $\beta$ CLS occupied and a single $\alpha$ occupied on top. The gap in this case is $V + 2\tau$, which vanishes for $V = -2\tau$. Along this critical line in the $\tau - V$ space, the ground state becomes $2^N$ times degenerate with all possible occupations of $\alpha$ modes on top of having all the $\beta$ already filled. As one decreases $V$ further below $-2\tau$, the new ground state becomes that of all $\alpha, \beta$ completely filled, and hence is unique. In this set-up without filling constraint, we name this (all filled state) the Exotic$_{\alpha\beta}$ phase. A similar phase transition happens for $\tau < 0$ as well. In this case, as one lowers $V$ below 0, the phase transition from Vanilla$_\alpha$ to the Exotic$_{\alpha\beta}$ phase occurs along the line $V = 2\tau$. The phase structure in the $\tau - V$ space, have been shown in Fig. (6), and ground states are tabulated in Table (2).

## 6.2 Carroll breaking deformation

Let us now consider the interacting analogue of the full free Hamiltonian for $\Delta \neq 0$ and choose $\Delta$ to be treated perturbatively, as we had done in the last section. So the total Hamiltonian again can be split into ultra-local $H_{\mathrm{ul}}$ and dispersive $H_{\mathrm{d}}$ parts:

$$H = H_{\mathrm{ul}} + H_{\mathrm{d}}. \tag{74}$$

Here $H_{\mathrm{ul}}$ is the ultralocal interacting Hamiltonian (33) and $H_{\mathrm{d}}$ is the second line of (28). Let us now revisit how the phase structure of $H_{\mathrm{ul}}$ behaves under this perturbation and, especially

for this subsection, focus at a point $0 > V > -2\tau, \tau > 0$, preferably near the blue-red critical line $V = -2\tau$.

**The vanilla phases: Unstable ultralocality**

The ground state in this phase is exactly the same as that of the half-filling vanilla phase. Although the excited part of the spectrum, in this case, is completely different than that of the half-filling case, the ground state itself, being half-filled, has non-zero matrix elements only with half-filled excited states. Hence, the first excited states don't contribute to perturbative correction to the ground state. Accordingly, the effect of perturbation to the ground state by the Carroll-breaking term is exactly as in (62). Similarly, The correlation functions cease to be strictly ultra-local, and a finite correlation length emerges, as verified in DMRG calculations Fig. (7a) and (7b).

**The exotic phase ground state: Robust ultralocality**

This phase is truly exotic in the sense that the ground state has all the $2N$ fermions the lattice can accommodate, filled. In terms of holes of $\alpha, \beta$, this ground state is a true vacuum. In addition to being a tensor product state of all the CLS modes, this is also a tensor product state in terms of site-local $c, d$ modes.[14] It follows directly that this state has zero entanglement. The correlations are as expected:

$$\langle c_i^\dagger c_j \rangle = \delta_{ij}, \qquad \langle d_i^\dagger d_j \rangle = \delta_{ij}, \qquad \langle c_i^\dagger d_j \rangle = 0. \tag{75}$$

From the correlation function itself, we can conclude that any quadratic hopping term annihilates this state. Hence, this state doesn't receive any perturbative correction under the influence of the dispersive Hamiltonian $H_d$. Therefore, the correlations for this phase are still ultra-local. This has been corroborated by DMRG-based numerical calculation, as described in Figure (7c) and (7d).

**Effect of perturbation on the gap**

This requires slightly more thought than what we found in the last section. Let us consider a point in the phase plot in the Vanilla$_\beta$ phase, preferably near the critical line that separates it from the Exotic$_{\alpha\beta}$ phase. For this set of parameters for $H_{\mathrm{ul}}$, the ground state has energy $\varepsilon = -2\tau N$, and the first excited subspace is spanned by $N$ degenerate states, each of energy $V + 2\tau + \varepsilon_0$, hence making the gap $V + 2\tau$. These states have all the $\beta$ modes filled up and just one $\alpha$ excited, as explained in the last section. The basis can be depicted as per our previous notation:

$$\left\{ |\psi_n\rangle = \left| \begin{array}{c} \overset{n}{\overbrace{\circ\circ\ldots\circ \, \bullet \, \circ\ldots\circ}} \\ \bullet\bullet\cdots\ldots\bullet\cdots\ldots\bullet \end{array} \right\rangle \middle| n = 1,\ldots,N \right\}. \tag{76}$$

The matrix elements of the perturbed Hamiltonian (60) in this basis of first excited subspace are:

$$\langle \psi_m | H | \psi_n \rangle = (V + 2\tau + \varepsilon_0)\,\delta_{m,n} + 2\Delta\left(\delta_{m,n+2} + \delta_{m,n-2}\right). \tag{77}$$

For large $N$, the eigenvalues can be approximated as

$$\varepsilon_n = V + 2\tau + \varepsilon_0 - 4\Delta \cos\left((n-1)\pi/N\right). \tag{78}$$

---

[14]This is because there is a unique vector in the highest form sub-space in the exterior algebra over a finite-dimensional vector space.

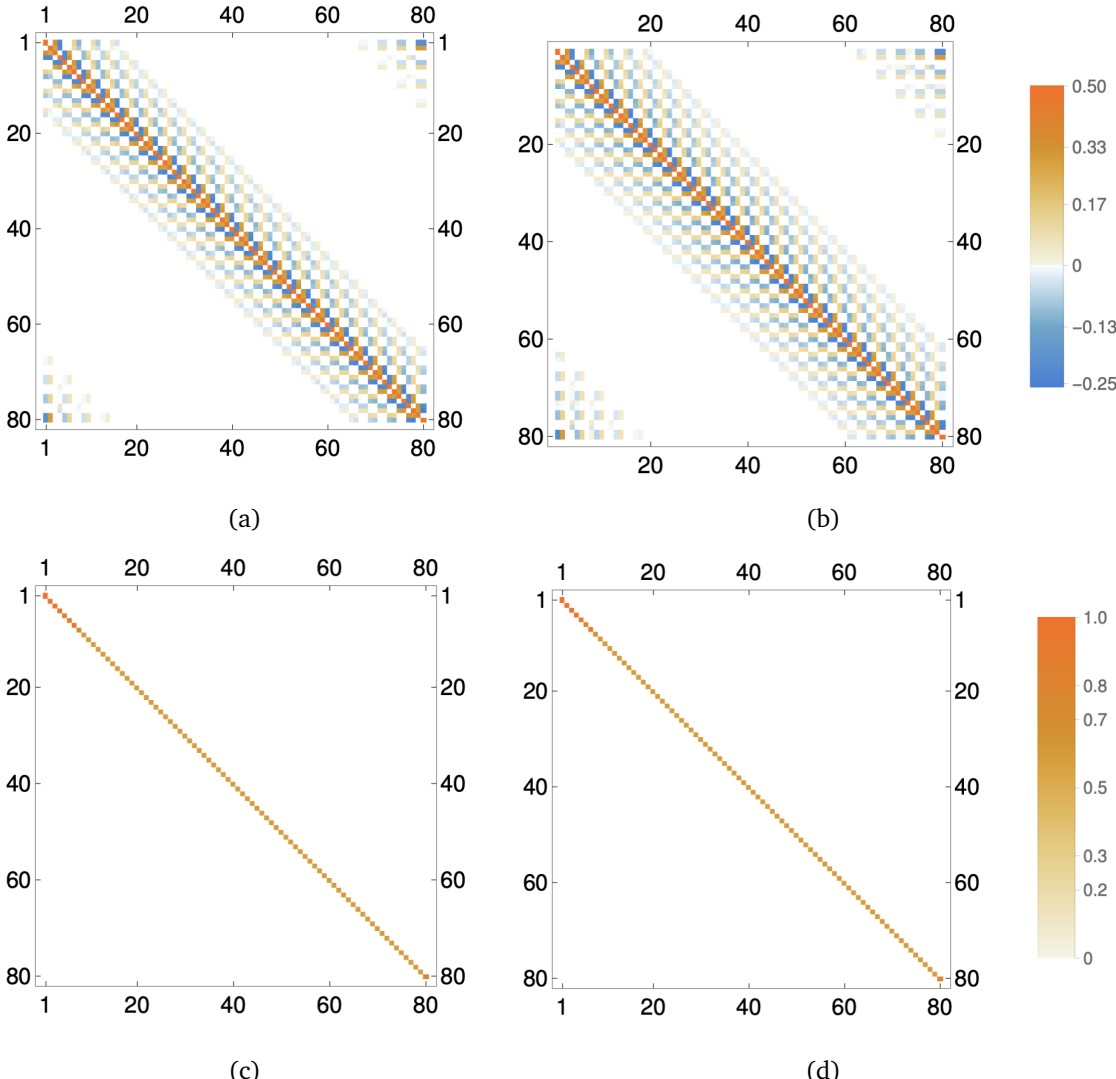

Figure 7: The effect of adding the Carroll breaking term (with $\Delta = 0.01, \tau = 0.4$) for the unconstrained filling case is illustrated in the matrix plots. If the critical point is approached from the right side of the blue line $V = -0.4$, (a) shows finite correlation length, increasing until the critical point $V = -0.8$ (b). Away from the critical line, deep inside the fully filled (Exotic) phase, we see the ultra-local feature for $V = -1.6$ in (c) and for $V = -2.0$ in (d). All plots here are for $N = 80$ sites with periodic boundary conditions.

We keep it noted that this result is non-perturbative in nature, as smallness in $\Delta$ has not yet been assumed in this section. The gap in this case gets perturbed to $V + 2\tau - 4\Delta$, as long as $\Delta$ is small to prevent level crossing. As one keeps $\tau, \Delta$ fixed and decreases $V$, towards $V + 2\tau - 4\Delta \to 0^+$, the gap gradually tends to zero. Unless in a purely Carrollian theory, here near the gap closure situation, there is a near-continuum set of states as per (78) and the correlation length increases, as per (63), and as explained above in (6.2), captured numerically in Figure (7b).

On the other hand, if one approaches criticality from the left side of the blue critical line, i.e. a point $V < -2\tau, \tau > 0$, a similar situation is encountered at the level of gap. The ground state is the one with all the $\alpha$ and the $\beta$ modes occupied, with the ground state energy being $\tilde{\varepsilon}_0 = NV$. The first excited states have an $N$-fold degeneracy for $\Delta = 0$, at an energy

$(N-1)V - 2\tau$. These are the states with only one $\alpha$ mode vacant. The gap, in this case, is $-V - 2\tau$. The first excited states' degeneracy is broken exactly similarly as in the other side of the phase, for $\Delta > 0$. For appropriately small $\Delta$, the gap gets modified to $-V - 2\tau - 4\Delta$. Hence, for fixed $\tau, \Delta$, the whole range $-2\tau - 4\Delta < V < -2\tau + 4\Delta$ gives a gapless phase.

Within this range, though, only the portion $-2\tau < V < -2\tau + 4\Delta$ gives a non-zero correlation length. Whereas for $-2\tau - 4\Delta < V < -2\tau$, one still has ultra-local correlations, despite being in a gapless phase, for reasons mentioned above in (6.2).

# 7 Discussions and conclusions

**Summary**

In this work, we uncovered a striking connection between the phenomena of compact localised states (CLS) and Carrollian symmetry. Although CLS are familiar in the field of condensed matter theory as a possible starting point for generating flat band systems, the space-time symmetry structure responsible for their occurrence has largely been obscured till now. Based on our earlier ideas regarding the flattening of energy bands and emergent supertranslation symmetries, we explicitly linked the imminent ultra-locality associated with CLS as well as supertranslation invariance in the relevant basis states. In our approach, there need not be a by-hand adjustment of hopping values to flatten the band, but we could intrinsically use Carrollian representation of fermions for the same.

Moreover, we could engineer interaction terms added to the pristine flat band theory out of the CLS basis states. Based on the strength of such interactions and the filling constraints of the states, one could see the phase structure in the parameter space becomes highly intriguing. Quantum Phase Transitions drive one to a phase with a highly degenerate ground state, where defining translation invariant states is hard. We circumnavigated the issue by introducing a symmetrization procedure, which preserves the manifest ultra-locality of the system. In this new phase, the ultra-locality turns out to be robust under Carroll-breaking transformations as well. Flat bands in usual settings are known to be extremely sensitive to perturbations, and most physically interesting perturbations move the theory away from ultra-locality. But from our symmetry considerations, it turns out one may actually be able to evade this in certain phases.

**The road ahead**

Now that we have understood various ideas pertaining to formalism, we believe this work would be the beginning of a series of thorough investigations in the structure of flat-bands, and in general, strongly correlated matter with emergent ultra-local features. Numerous questions remain, and let us give the reader a glimpse of a few.

*The continuum limit:*

In this work, we introduced a lattice version of the ultra-local fermions with interactions. Although our exotic phase does share some structures with the free Carroll theory, it is not clear apriori whether one can take our interacting model (33) to a continuum limit. As discussed previously, the coupling strength $V$ is marginal in the continuum limit, while $\tau$ is relevant. This poses a challenge in understanding this limit as connected to the continuum of Carroll fermions away from the gapless region.

*Finite entanglement entropy:*

As observed in this work, as well as in other explicit examples [57, 80, 81, 93] of strictly ultra-local (trivial CLS) Carrollian theories, entanglement entropy seems to be trivial, i.e. independent of the size of the equal-time subsystem. However, holographic computations [33, 94] in the presence of gravitational anomaly predict the possibility of non-trivial scaling of entanglement entropy for conformal Carrollian theories. In this sense, working with an explicit example that is not strictly ultra-local (like the ones known as the magnetic Carroll theories [72]) theory by nature would be a good exercise to perform towards this direction.

*Other information theoretic markers:*

Since we have a non-trivial phase transition associated with our interacting system, it is a valid question to ask whether other quantifiers than entanglement can give us more information about such an exotic phase. One could, in principle, ask for a precise measure of the complexity of a state on two sides of the phase boundary. Effectively, that can be computed via Nielsen's method [95] where discontinuities may be able to detect phase transitions. In recent times other classes of complexity measures, such as Krylov (or spread) complexity [96] has been touted to be sensitive to topological phase transitions (see [97] for example). It might be instructive to compute such markers for our interacting system to clear up the nature of the phases. It would be potentially significant to investigate the nature of quantum complexity across our gapless lines if the final state is highly degenerate, as in the exotic phase.

*Emergent supertranslations in lattice systems:*

As we have been clearly noticing, the emergent supertranslation symmetries pop up in a myriad of physical situations. It is highly likely one may be able to understand some so-called ill-understood exotic lattice phenomena under the aegis of such a formulation. In recent advancements, the instability produced at the large compressibility limit of low energy theories with a Tomonaga-Luttinger description has been described as having an effective Carrollian description [98]. We hope to identify more such exquisite connections in future projects.

*Flat holography/CMT:*

The present work is a part of a greater body of work aiming at understanding fermionic systems (possibly even including supersymmetry [99]) with Carrollian symmetries in light of a putative holographic prescription for gravity in asymptotically flat space-time. Preliminary checks in this direction, including flat dispersion, correlation functions, triviality of entanglement and global charge algebra, point towards a robust framework comparable to that of the study of condensed matter systems dual to gravity in AdS spacetime [100]. A coordinated approach towards such a realisation would be a powerful step to take.

We have only scratched the surface of the rich physics of ultra-local supertranslation invariant systems in this work, and we surely plan to report on some of the above directions in future communications.

## Acknowledgments

The authors would like to thank Arjun Bagchi, Yunkyu Bang, Daniel Grumiller, Indrakshi Ray-chowdhury, Anna Kauch, Arijit Kundu, Emil Mathew, and Sourish Banerjee for many insightful discussions.

**Funding information** ABan is supported in part by an OPERA grant and a seed grant NFSG/PIL/2023/P3816 from BITS Pilani, and further an early career research grant ANRF/ECRG/2024/002604/PMS from ANRF India. He also acknowledges financial support from the Asia Pacific Center for Theoretical Physics (APCTP) via an Associate Fellowship. Part of the results in this work were presented in the APCTP/GIST workshop "New Avenues in Quantum Many-body Physics and Holography" in December 2024. The grants that support the research of RB are CRG/2020/002035, MTR/ 2022/000795 from SERB, India, and CDRF and OPERA grant from BITS Pilani. NA and RB thank the Indo-Austria bilateral research grant DST/IC/Austria/P-9/2021. NA would like to thank the Start-up Research Grant (SRG/2022/000972) for the computational facilities, OeAD and TU Wien for their hospitality. The authors further acknowledge Sharanga HPC, BITS Pilani central facility usage for tensor network computations.

# A Criticality and the BMS$_3$ symmetry

An infinite dimensional Lie algebra of vector fields generates asymptotic symmetries of 2+1 dimensional Einstein gravity in asymptotically flat space-time:

$$\left[L_{f_1}, L_{f_2}\right] = L_{f_1 f_2' - f_1' f_2}, \qquad \left[L_f, M_g\right] = M_{f g' - f' g}, \qquad \left[M_{f_1}, M_{f_2}\right] = 0. \qquad (A.1)$$

In terms of the local coordinate $(t, x)$ patch of the asymptotic null infinity $\mathcal{I}^+$, $f, g$ etc. are arbitrary functions of the spatial coordinate $x$ and the vector fields themselves are:

$$L_f = f(x)\partial_x + f'(x)t\partial_t, \qquad M_f = f(x)\partial_t. \qquad (A.2)$$

This goes by the name of BMS$_3$ [29] algebra or the conformal Carroll algebra in 2 space-time dimensions [27]. Just as the Virasoro algebra generates space-time symmetries in 1+1 dimensional relativistic conformal field theories, the BMS algebra plays the same role for ultra-relativistic/ Carrollian conformal field theories. The later ones may be realized as the speed of light, $c \to 0$ limit of the relativistic case. One readily identifies the class of vector fields $M_f$ to be equivalent to supertranslation generators we introduced in (7).

Just as any 1+1 dimensional relativistic theory at the gapless limit enjoys the symmetries generated by Virasoro algebra, we may expect that the flat-band models or their interacting counter-parts, e.g. (33) should have the full BMS symmetry (A.1) at the critical points. It is instructive to note that $L_{f(x)=x} = x\partial_x + t\partial_t$ is the generator of scaling symmetry, whose form is the same as that of a relativistic conformal field theory. It is natural to expect that scale invariance calls for additional symmetries in the supertranslation invariant theories. Only a finite number of them, though, are required to constrain the lower point functions without detailed knowledge of the Hamiltonian. For example, the form (41) of the 2-point function can be obtained by solving the Ward identities corresponding time translation: $M_0 \equiv M_{f = \text{const}}$, Carroll boost: $M_1 \equiv M_{f(x)=x}$, spatial translation: $L_0 \equiv L_{f = \text{const}}$ and dilatation: $L_1 \equiv L_{f(x)=x}$. To see the emergence of the form (41), we define the following transformation rules:

$$\begin{aligned}
\delta_{M_0} O_\Delta(t, x) &= \partial_t O_\Delta(t, x), & \delta_{L_0} O_\Delta(t, x) &= \partial_x O_\Delta(t, x), \\
\delta_{M_1} O_\Delta(t, x) &= x\partial_t O_\Delta(t, x), & \delta_{L_1} O_\Delta(t, x) &= (t\partial_t + x\partial_x + \Delta)O_\Delta(t, x).
\end{aligned} \qquad (A.3)$$

Note that the above transformation rules are different than the choice of representation traditionally made in 2D Carrollian CFT [101, 102], and is inspired by the representation used in the higher dimensional cousin [36], in a sense that the weight associated with the action of $M_0$ is chosen to zero here. A natural choice of spatially ultra-local two-point function

$G_{\Delta_1, \Delta_2}(t_1, x_1; t_2, x_2) = g(t_1 - t_2)\delta(x_1 - x_2)$ does satisfy the Ward identities of $M_0, L_0, M_1$. The Ward identity of $L_1$ leads to the differential equation:

$$t\partial_t g(t) + (\Delta_1 + \Delta_2 - 1)\, g(t) = 0\,. \tag{A.4}$$

This solves for $g(t) = C_{\Delta_1, \Delta_2} t^{(1 - \Delta_1 - \Delta_2)}$ and hence (41) follows. Other generators don't impose any further constraints to determine $C$.

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
