# Peer review of "Flat Bands and Compact Localised States: A Carrollian roadmap"

_SciPost Physics, doi:SciPost Phys. 19, 046 (2025)_

## Round 2 · Referee Report · Anonymous (Referee 1) · 2025-3-10

Report

In this work, the authors studied how to use the Carrollian symmetry to construct one-dimensional lattice fermionic systems with all flat-band spectra. They discovered a striking relation between the compact localized states (CLS) and Carrollian symmetry. They showed that the CLS modes, which are essential in generating flat-band system, can be generated by the supertranslation transformation in Carrollian quantum field theory. With the site-local CLS modes, they were allowed to construct the supertranslation invariant interaction terms. In the work, they introduced four-fermion interactions into the systems and studied the quantum phase structure, both with the filling factor constrained at half and unconstrained.

The work presents some novel and important findings. It may lead to further studies of the strongly correlated systems with emergent ultra-local features from Carrollian symmetry point of view. I would like to recommend it for publication in its present form.

Recommendation

Publish (easily meets expectations and criteria for this Journal; among top 50%)

---

## Round 2 · Referee Report · Anonymous (Referee 3) · 2025-6-12

Report

The work by Ara, Banerjee, Basu and Krishnan is very interesting and deserves in my opinion publication in a high-level journal such as SciPost. It easily meets expectations and criteria for this journal, as fas as I can tell.

In my opinion, one high added value of this work is to establish and elaborate on a new connexion between two seemingly disparate communities: that of high-energy/gravitational physics and condensed matter theory. Common grounds between these two fields are of course not new, and date back to 2007-2008 with the first occurrences of AdS/CMT. The present paper however discusses a new and potentially far-reaching occurrence of this connexion: the relevance of a particular type of symmetry (Carroll), that was argued to appear in flat band systems in particular. This symmetry has nowadays become instrumental in the endeavour of understanding gravitational systems in asymptotically flat space-times under the name of "Carroll holography" and its links to the celestial holography program.

The authors present a construction of flat-band fermionic lattice systems in one spatial dimension, characterized by Carrollian symmetry and supertranslation invariance. From nilpotent matrices, they construct ultra-local Hamiltonians whose eigenstates are Compact Localised States (CLSs). The paper further explores the dynamical and entanglement properties of such systems, introduces supertranslation-invariant interactions, and analyses quantum phase structure (both at and away from half-filling), highlighting an exotic phase with a highly degenerate ground state.

High-energy/gravitational physicists might not be not well-acquainted with some of the condensed-matter concepts the paper deals with, which could result in requiring efforts to get into some parts of the paper, which is otherwise very well-presented. Since I believe the results of this paper could be of high interest for both high-energy and cond-mat communities, and in order to bridge the gap more smoothly between them making the paper even more useful, I would suggest to elaborate on some aspects and material tackled in the paper and not necessarily background material for the gravity community.

Below is listed (in order of appearance in the manuscript) a set of questions/remarks of various levels of importance, from trivial to more technical, and from typos to general clarifications. I believe it would be useful if the authors could address these points.

Comments/questions:

(1) p2: Could you be more precise about what is meant by "playing around with some form of local symmetries."?

(2) p2: "Despite this, a clear and universal Lie algebra-based understanding of it has been scarce.". Which Lie algebra?

(3) p3: "resultant" -> resulting?

(4) p3: "Nevertheless, it has been shown conclusively that infinite Carrollian supertranslation invariance of the Hamiltonian and flat bands is intricately related to each other [20]." -> is a bit redundant with last paragraph pf page 2.

(5) p5: What is the mathematical definition of CLSs? Is it (2.7)? The fact the Hamiltonian tales the form (2.9)?

(6) p5: How does (2.7) relate to the "gravity" definition of supertranslations, i.e. angle-dependent translations? (in App. A for instance)

(7) p8, bottom. Could you point at the precise relation between the Schrodinger and Carroll algebras (either explicitly in the text or reference)?

(8) p9, paragraph 1 "For more details on the interplay between energy scales and symmetry groups for Carroll invariant theories". What is the relation between taking the $t_1 \right arrow t_2$ limit and energy scales?

(9) p9: "can be intrinsically generated using the nilpotent matrices (2.1), making our construction of flat dispersion models simple yet profound.". I agree with that statement. Is this the first time that the connexion between flat band systems and nilpotent matrices is established?

(10) p11: "the symmetry group including conformal transformation, is generated by the BMS3 algebra [28]:" Is [28] the fist occurrence of the BMS3 algebra? See e.g. Barnich-Compère (2006) (and references therein).

(11) p11-12: That part of the paper refers to the Conformal Carroll algebra in 2d, while before it was only question of the Carroll algebra (including supertranslations). Do the models (2.1)-(2.12) enjoy superrotations?

(12) p11-12: If no, to the previous question, why is (3.9) relevant? Can it be derived for a 2d Carroll field theory (not a Conformal Carroll field theory)?

(13) p15: is (3.18) exact, or only to quadratic order in $\Delta/\tau$? Is this result universal in (C)Carroll FTs? Is there a gravity/holographic counterpart of such an expression? (e.g. poles of a thermal Green's function?)

(14) p16: Please define scar-like states. Is the claim that translation-symmetrized CLSs behave like quantum scar states? Aren't these usually associated with non-integrable systems?

(15) p18: notation for $C_{N/2}$?

(16) p19: How is scaling dimension defined? With respect to which generator of the Carroll algebra? (see also (11))

(17) p20: "to understand how our eigenstates written " -> are written

(18) p21: is [53] the first time (5.12) was written?

(19) apriori -> a priori

(20) Sect 5.4. Make explicit the trace resulting in (5.34)?

(21) Sect 5.4.: From (5.34), can anything be said about the entropy of the system? Does it compare to (some limit of) (3.9)?

(22) As mentioned in the conclusion, ``(this work) point(s) towards a robust framework comparable to that of the study of condensed matter systems dual to gravity in AdS spacetime". In the context of AdS/CFT, an intriguing connexion between AdS gravity and the canonical Ising model has been suggested in 1111.1987, in particular that the partition function of pure Einstein gravity with $c=1$ matches that of the Ising model. Is there a ``flat limit" version of this statement, and would it relate to the ultra-local model addressed in this paper?

(23) Define some concepts, or point at references: topological phases, DMRG, fidelity,...

Recommendation

Ask for minor revision

---

## Round 2 · Referee Report · Anonymous (Referee 2) · 2025-6-12

Strengths

1- Find applications of exotic spacetime symmetries to condensed matter systems 2- Specifically, identify physical property of compact localized states with the mathematical property of supertranslation invariance, a key aspect of Carollian symmetries 3- Explicitly provide interacting theory with non-trivial phase structure with the correct symmetries 4- Discuss numerous consequences and properties of this theory

Weaknesses

1- The Appendix A misses the target audience and contains confusing statements. 2- The formula (3.9) for entanglement entropy is not quite correct. It misses an additive term depending on the cutoff ratio (that was also missed in [32]). If for some reason this term is not relevant the authors should mention why; otherwise, they should discuss it. 3- It is unclear if the continuum limit of their interacting model (4.1) exists.

Report

The paper easily meets the journal's acceptance criteria, as it strengthens the remarkable link between seemingly exotic spacetime symmetries - Carroll symmetries, which arise in the limit of vanishing speed of light from Poincare - and condensed matter systems with specific phase structure, esepcially the emergence of flat bands. It is not trivial to find interacting theories that exhibit Carrollian symmetries and the model (4.1) introduced and studied in this paper constitutes important progress. Given the listed strengths and the (by comparison) marginal weaknesses I suggest publication of this work in SciPost Physics after the authors amended their manuscript to address the weaknesses.

Requested changes

1- There is a repeated typo where they write "Appendix.(A)". The "." and parentheses should be removed. 2- When they mention holographic computations of entanglement entropy they probably should add the holographic computation of entanglement entropy by W. Song et al, e.g., the one using swing surfaces. 3- I do not think Appendix A reaches the target audience. In particular, the algebra (A.1) does not look at all like the algebra (3.8) in the main text, so they should add an explanation how introducing modes in (A.1) produces (3.8). Another subtlety that they do not explain in Appendix A is how the central charges emerge. Their vector field representation (A.2) leads to c_L=0=c_M, which is not what they are using in the main text. I found also the quote of Ref. [28] misplaced when mentioning BMS_3. Instead, they should quote gr-qc/9608042 and gr-qc/0610130 where this algebra was introduced (in the second case with central extensions). 4- Given that one of the key points of the papers is to come up with an interacting model that in some limit has Carroll symmetries it seems fair to mention other constructions of interacting theories with Carroll symmetries, for instance, models with spacetime subsystem symmetries, see 2303.15590 (while the relation to Carroll was not worked out so clearly in that paper, some of the later papers made this connection transparent and could be quoted as well).

Recommendation

Publish (surpasses expectations and criteria for this Journal; among top 10%)

---

## Round 3 · Author Response

Warnings issued while processing user-supplied markup:

  • Inconsistency: Markdown and reStructuredText syntaxes are mixed. Markdown will be used.
    Add "#coerce:reST" or "#coerce:plain" as the first line of your text to force reStructuredText or no markup.
    You may also contact the helpdesk if the formatting is incorrect and you are unable to edit your text.

Dear Editor,

We thank the referee for meticulously going through the manuscript and for their insightful comments, which have definitely contributed to improving the quality of the work. Below we address the comments and suggestions made by the referee point by point. Associated changes in the revised manuscript have been put in at relevant places.

Reply to Report #4 by Anonymous (Referee 3)

  1. p2: Could you be more precise about what is meant by "playing around with some form of local symmetries."?

    Reply: We have replaced this by a clearer statement "At the level of single particle spectrum, flat bands imply the existence of a large number of degenerate states, and hence a large number of generators of, possibly continuous, symmetries."

  2. p2: "Despite this, a clear and universal Lie algebra-based understanding of it has been scarce.". Which Lie algebra?

    Reply: We have rephrased this as "Despite this, a systematic study of these symmetries in terms of Lie groups/ algebras has been scarce"

  3. p3: "resultant" $\rightarrow$ resulting?

    Reply: We have made the correction in the manuscript.

  4. p3: "Nevertheless, it has been shown conclusively that infinite Carrollian supertranslation invariance of the Hamiltonian and flat bands is intricately related to each other [20]." $\rightarrow$ is a bit redundant with last paragraph of page 2.

    Reply: We have omitted this statement.

  5. p5: What is the mathematical definition of CLSs? Is it (2.7)? The fact the Hamiltonian tales the form (2.9)?

    Reply: In this case, yes, eq. (2.7) is the definition of the CLS. At any rate eq. (2.9) is a form of any single particle state. What sets the CLSs apart is that the quadratic form eq. (2.9) can be arrived at without going to a Bloch wavefunction basis, and the support of $\xi_j$ over the spatial lattice is compact.

  6. p5: How does (2.7) relate to the "gravity" definition of supertranslations, i.e. angle-dependent translations? (in App. A for instance)

    Reply: Please refer to the first equality of eq. (2.7) and compare with the definition of supertranslation $M_f$ in eq. (A2). For gravity in asymptotically flat space-time, these $M_f$ generators define symmetries of the asymptotic boundary, which has cylindrical topology. Here $x$ coordinatizes points, as angles, on the circular foliations, whereas $t$ refers to the null-time of the boundary. Hence, the supertranslations are also named angle-dependent time translation. More often than not, in order to understand quantities in a holographic renormalization flow set-up, one needs to put a lattice cut-off on the circle, making $x$ discrete. In that case, one gets an exact equivalence between the definition of the transformation generator $\delta_f$ of (2.7) and $M_f$ of (A.2).

  7. p8, bottom. Could you point at the precise relation between the Schrodinger and Carroll algebras (either explicitly in the text or reference)?

    Reply: We have added a couple of references, where these relations are spelled out, towards the end of the last paragraph of page 8.

  8. p9, paragraph 1 "For more details on the interplay between energy scales and symmetry groups for Carroll invariant theories". What is the relation between taking the $t_1 \rightarrow t_2$ limit and energy scales?

    Reply: We have moved this statement toward the end of the penultimate paragraph of page 8, and for clarity, elaborated more on the connection between merging energy scales (UV/IR mixing) and the $t_1 \rightarrow t_2$ limit.

  9. p9: "can be intrinsically generated using the nilpotent matrices (2.1), making our construction of flat dispersion models simple yet profound.". I agree with that statement. Is this the first time that the connection between flat band systems and nilpotent matrices is established?

    Reply: Yes, to the best of our knowledge, our work is the first to generate flat band models using a nilpotent matrix.

  10. p11: "the symmetry group including conformal transformation, is generated by the BMS3 algebra [28]:" Is [28] the fist occurrence of the BMS3 algebra? See e.g. Barnich-Compère (2006) (and references therein). Reply:* We thank the referee for mentioning this. The reference to Barnich-Compère has been added. The explicit form of BMS$_3$ Lie algebra probably appeared in their work first.

  11. (11) p11-12: That part of the paper refers to the Conformal Carroll algebra in 2d, while before it was only question of the Carroll algebra (including supertranslations). Do the models (2.1)-(2.12) enjoy superrotations?

    Reply: The $L_n$ generators of BMS$_3$ are not symmetries of the models discussed. This is simply because these models are gapped, whereas $L_0$, for example, is the dilatation generator. We alluded to the discussion of conformal Carroll, for drawing a similarity with the behaviours of entanglement entropy in Carroll symmetric models like ours. This points toward the fact that subsystem size independence of entanglement entropy in Carrollian theories seems to be a feature, irrespective of conformal symmetry. A side-remark: the single copy of Witt algebra generators, ie. $L_n$ of BMS$_3$ aren't strictly superrotations, as superrotations are there in 2+1 or higher dimensions.

  12. p11-12: If no, to the previous question, why is (3.9) relevant? Can it be derived for a 2d Carroll field theory (not a Conformal Carroll field theory)?

    Reply: Partially answered in the above point. Moreover, when we later introduce interactions, there are phase transitions, where the system does have scale invariance. As a prelude to that, we found it useful to keep the referring to conformal Carrollian theories.

  13. p15: is (3.18) exact, or only to quadratic order in $\Delta/\tau$? Is this result universal in (C)Carroll FTs? Is there a gravity/holographic counterpart of such an expression? (e.g. poles of a thermal Green's function?)

    Reply: No, the result is not exact, and was found by making a $\Delta/\tau$ expansion in the integrated up to linear order in perturbation. To the best of our understanding, the exact form of the return probability depends upon the Carroll breaking deformation term. The holographic picture of this involves deforming bulk flat space gravity by a small cosmological constant. In fact, as per our previous finding, local Carroll breaking deformations are relevant under an RG flow. Hence terms like these, however perturbatively small in the boundary theory, would signify emergence of AdS geometry deeper in the bulk. As correctly pointed out by the referee, the effect of this (either in return probability or a linear response coefficient) would indeed show up in 2 point function. These studies are very much work in progress, and would be reported elsewhere.

  14. p16: Please define scar-like states. Is the claim that translation-symmetrized CLSs behave like quantum scar states? Aren't these usually associated with non-integrable systems?

    Reply: We made this remark, keeping in mind the subsystem size independence of entanglent entropy, which is one of the defining properties of a scar state. In fact scar states in flat band systems have already been studied in detail. See for example, the references mentioned in the footnote of page 16.

  15. p18: notation for $C_{N/2}$?

    Reply: Probably the referee is pointing toward ${}^N C_{N/2}$. This is standard notation for "$N$ choose $N/2$".

  16. p19: How is scaling dimension defined? With respect to which generator of the Carroll algebra? (see also (11))

    Reply: Scaling dimension is defined with respect to $L_0$, ie. dilatation of the conformal Carroll algebra. As mentioned above, at the phase transition points (section 5.1), the gap closes, and hence it is natural that scale invariance emerges in the system.

  17. p20: "to understand how our eigenstates written " $\rightarrow$ are written

    Reply: The full line now reads: "First let us try to understand how our eigenstates, written in the CLS basis, work."

  18. p21: is [53] the first time (5.12) was written?

    Reply: This is too trivial a theory to be studied from the perspective of QFT, a subject that historically was developed to address questions in high energy scattering phenomena. However, one of the present authors studied this in ref. [37] as well, again for scattering amplitudes, but from a boundary field theory perspective. Hence we included this reference here. To the best of our knowledge we didn't encounter this action, in this form, previous to these references.

  19. apriori $\rightarrow$ a priori

    Reply: We have made the correction in the manuscript.

  20. Sect 5.4. Make explicit the trace resulting in (5.34)?

    Reply: This is basic statistical mechanics, where each degree of freedom has the same spectrum. In this case one can construct the full grand canonical partition function using the single particle partition function. The reader may refer to any standard book like Kerson Huang for this.

  21. (21) Sect 5.4.: From (5.34), can anything be said about the entropy of the system? Does it compare to (some limit of) (3.9)?

    Reply: Our main motivation behind studying the thermodynamics in Sec 5.4 was to capture the phase transition at the critical point $V = -4\tau$. In calculation of the average energy density it is not differentiable with respect to temperature across the critical point, clearly indicating the phase transition. Hence we didn't need to study any other thermodynamic property like entropy. However, one can easily see from the structure of the grand canonical partition function that the thermodynamic entropy should scale linearly with system size, $N.$ On the other hand, entanglement entropy at the ground state doesn't scale with subsystem size. Hence, we don't think of a direct way to compare the two.

  22. As mentioned in the conclusion, (this work) point(s) towards a robust framework comparable to that of the study of condensed matter systems dual to gravity in AdS spacetime". In the context of AdS/CFT, an intriguing connexion between AdS gravity and the canonical Ising model has been suggested in 1111.1987, in particular that the partition function of pure Einstein gravity with $c=1$ matches that of the Ising model. Is there a flat limit" version of this statement, and would it relate to the ultra-local model addressed in this paper?

    Reply: 1111.1987 proposed an equivalence based on mapping states on gravity to states on the $c=1$ minimal model on the CFT side. This, in our opinion, is a rare case where a Hilbert space interpretation in highly quantum regime $l/G \sim 1$ of gravity is possible. There has been works towards such attempts for asymptotically flat gravity. See for example, 2307.00043, a rather recent work, where the initial steps towards Hilbert space interpretation from gravity partition function through modularity properties have been studied.

    However, in the present work and it's future follow-ups we are rather interested in RG flow mechanism from the holographic set-up. For this precise reason, we discussed the role of energy scales in determining Carrollian symmetry.

  23. Define some concepts, or point at references: topological phases, DMRG, fidelity,...

    Reply: These are very standard terms which have been used generally in Physics literature for quite a long time. For the sake of the reader, who may not be well versed in these topics, we have added a few references for related reviews.

We also take this opportunity to thank the editor for pointing out the reference JHEP07(2017)142, which we have included in our discussion on flat holography, in the introduction.

We have further taken care of small typos in the manuscript and added a few references missed earlier.

Now with all the comments and suggestions made by the referee (report 4) taken care of, we hope that the revised version of the manuscript can be accepted for publication without any further delay.

Best regards

Aritra (For the authors)

---

## Round 3 · List of Changes

Included in the resubmission letter.

---

## Editorial Decision

published